# Adaptive Conformal Anomaly Detection with Time Series Foundation Models for Signal Monitoring

**Natalia Martinez Gil**    **Fearghal O'Donncha**    **Wesley M. Gifford**    **Nianjun Zhou**
**Dhaval C. Patel**    **Roman Vaculin**

IBM Research
natalia.martinez.gil@ibm.com

## Abstract

We propose a post-hoc adaptive conformal anomaly detection method for monitoring time series that leverages predictions from pre-trained foundation models without requiring additional fine-tuning. Our method yields an interpretable anomaly score directly interpretable as a false alarm rate (p-value), facilitating transparent and actionable decision-making. It employs weighted quantile conformal prediction bounds and adaptively learns optimal weighting parameters from past predictions, enabling calibration under distribution shifts and stable false alarm control, while preserving out-of-sample guarantees. As a model-agnostic solution, it integrates seamlessly with foundation models and supports rapid deployment in resource-constrained environments. This approach addresses key industrial challenges such as limited data availability, lack of training expertise, and the need for immediate inference, while taking advantage of the growing accessibility of time series foundation models. Experiments on both synthetic and real-world datasets show that the proposed approach delivers strong performance, combining simplicity, interpretability, robustness, and adaptivity. [1]

## 1 Introduction

A common challenge in industrial applications such as predictive maintenance and signal monitoring is the scarcity of sufficient quality data and infrastructure to train robust models Cook et al. (2019); Ajami & Daneshvar (2012); Kanawaday & Sane (2017); Beghi et al. (2016); Shah & Tiwari (2018); Moghaddass & Wang (2017). This limitation can hinder the ability to make accurate and reliable predictions, which are essential to detect anomalies and ensure operational efficiency. Foundation models, particularly in the time series domain Liang et al. (2024), offer a promising solution. These models excel at leveraging prior knowledge and historical observations, enabling them to provide good enough initial estimates of expected values and statistical characteristics of monitored signals, even in data-scarce environments. This capability is invaluable for industries aiming to enhance their monitoring systems without the need for extensive datasets.

In the context of time series anomaly detection, an adaptive approach is crucial for monitoring and maintaining the reliability of signals. Anomalies, or deviations from expected behavior, can manifest in different forms, such as point anomalies, where an individual observation significantly deviates from normal patterns, and contextual anomalies, where a value is only considered anomalous within a specific temporal context Boniol et al. (2024). Detecting these effectively requires models that capture underlying temporal dependencies and adapt to non-stationary data distributions.

A prominent class of anomaly detection methods relies on predictive modeling, where a forecasting model learns normal time series behavior, and deviations between predicted and actual values could indicate anomalies in operations or shifts in operational modes that require expert attention Basseville (1993); Choudhary et al. (2017); Gama et al. (2014); Saurav et al. (2018). However, many existing approaches assume access to large amounts of training data, making them impractical in settings where only a few samples are initially available. This motivates the use of pretrained

---

[1]Code: https://github.com/ibm-granite/granite-tsfm/tree/main/notebooks/hfdemo/adaptive_conformal_tsad

Time Series Foundation Models (TSFMs) Rasul et al. (2023; 2024); Ansari et al. (2024); Liang et al. (2024), which have been trained on large-scale datasets and can generalize to new time series with minimal adaptation. Furthermore, existing anomaly detection systems often lack interpretability, relying on thresholding mechanisms that assume a fixed data distribution Schmidl et al. (2022); Paparrizos et al. (2022b); Goswami et al. (2022), which limits their adaptability to evolving time series data. In this setting, a robust system must balance sensitivity and adaptability, minimizing false alarms while effectively detecting significant behavioral transitions. This ensures timely identification of suspicious patterns without overwhelming experts with noise, fostering a more efficient and reliable monitoring framework Cook et al. (2019).

To address these limitations, we propose a conformal-based anomaly detection method that integrates the predictions of pretrained TSFMs with conformal prediction techniques Vovk et al. (2005); Angelopoulos & Bates (2021) to produce an interpretable, adaptive anomaly score directly linked to a desired alarm rate. Conformal methods offer model-agnostic and distribution-free uncertainty quantification with finite-sample guarantees, making them highly suitable for real-world anomaly detection. However, standard conformal approaches rely on the assumption of exchangeability, which is often violated in time series due to temporal dependencies. Furthermore, existing conformal methods for anomaly detection primarily focus on thresholding arbitrary anomaly scores derived from non-anomalous data while assuming exchangeability Angelopoulos & Bates (2021); Guan (2019); Bates et al. (2023), limiting their applicability in dynamic, non-stationary settings.

**Main Contributions** We propose $\mathcal{W}_1$-ACAS, a post-hoc adaptive conformal anomaly detection framework that leverages predictions from pretrained forecasters (e.g., TSFMs) to monitor signals without requiring fine-tuning. This is particularly valuable in industrial settings, where users often lack sufficient data, data-cleaning pipelines, or specialized expertise Cook et al. (2019). Our approach provides a practical solution for immediate anomaly monitoring. Figure 1 illustrates the method: (a) anomaly scores are derived as conformal $p$-values from forecaster errors across multiple horizons and aggregated into a single score; (b) anomalies are flagged when adaptive $p$-values fall below a threshold on real signals; and (c) the learned adaptive weights emphasize past errors with similar distributions, capturing recurring patterns such as periodicity, thereby improving detection while offering direct control over the alarm rate. Our framework offers the following properties:

- **Interpretability:** The anomaly score corresponds directly to an alarm rate ($p$-value), providing a transparent and probabilistic basis for decisions.

- **Distribution-Agnostic:** Built on quantile conformal prediction, the method is robust to heavy-tailed and complex error distributions.

- **Adaptivity:** By weighting past nonconformity scores via the Wasserstein distance, the framework adapts online to distribution shifts, reducing false alarms while preserving calibration Barber et al. (2023).

- **Post-Hoc and Model-Agnostic:** The method applies directly to pretrained TSFMs or any anomaly score, requiring no retraining while inheriting the guarantees of weighted conformal prediction. Its effectiveness is proved through integration with TSFM forecasters.

## 2 RELATED WORK

**Time Series Anomaly Detection** Prediction-based methods detect anomalies by comparing observed values against forecasts (Giannoni et al., 2018; Boniol et al., 2024). Recent TSFMs (Rasul et al., 2023; 2024; Ansari et al., 2024; Liang et al., 2024) are well suited for online detection in data-scarce scenarios, offering accurate zero-shot forecasting performance. Recent benchmark studies (Paparrizos et al., 2022b; Liu & Paparrizos, 2024) show that classical distance- and density-based methods (Li et al., 2007; Ramaswamy et al., 2000; Aggarwal & Aggarwal, 2017; Paparrizos & Gravano, 2015; 2017; Boniol et al., 2021) often outperform more complex models, but they typically require access to the full dataset (non-causal), lack robustness across temporal patterns, and are unsuitable for streaming settings. Moreover, many anomaly scores lack clear probabilistic meaning, and common thresholding strategies rely on full-dataset statistics (Ahmad et al., 2017), limiting real-time applicability. In practice, anomaly detection systems must not only achieve high accuracy but also provide interpretable confidence scores while maintaining low false alarm rates (Cook et al., 2019). Our work addresses these challenges by combining TSFMs with adaptive conformal scoring, yielding interpretable and calibrated thresholds for reliable streaming anomaly detection.

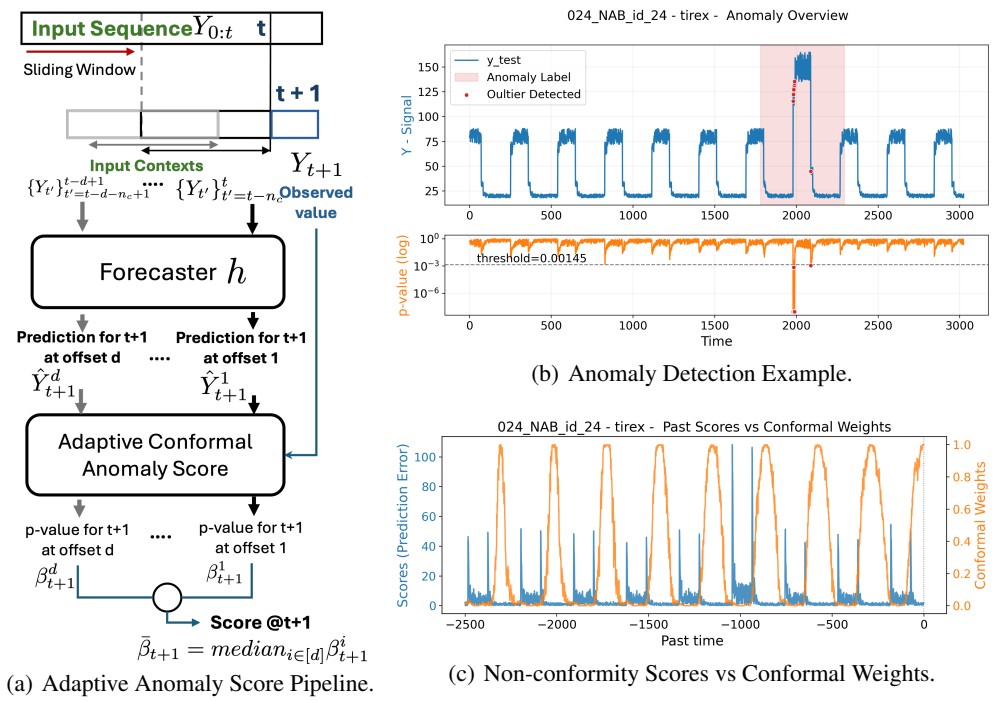

Figure 1: Illustration of our proposed $\mathcal{W}_1$-ACAS method. (a) Anomaly scoring pipeline: conformal $p$-values are computed across forecast horizons from forecaster errors and aggregated. The mapping is adapted online by weighting past nonconformity scores, with weights evolving to capture distributional shifts or recurring patterns. (b) Example signal (blue) with ground-truth anomaly labels, where detected outliers (red dots) occur when adaptive $p$-values (orange) fall below a threshold. (c) Converged adaptive weights (orange) over past errors (blue), averaged across horizons, shows how $\mathcal{W}_1$-ACAS captures error patterns with similar distributions, here reflecting its periodic behavior.

**Conformal Prediction.** Conformal prediction provides distribution-free uncertainty quantification with finite-sample guarantees (Vovk et al., 2005; Shafer & Vovk, 2008; Angelopoulos & Bates, 2021). A widely used variant, split conformal prediction (SCP) (Papadopoulos et al., 2002), is post-hoc and model-agnostic, relying only on model predictions and a calibration set. While effective under exchangeability [2], this assumption is often violated in time series settings, motivating adaptive extensions. Recent works (Gibbs & Candes, 2021; Zaffran et al., 2022; Gibbs & Candès, 2024) adjust conformal quantiles online to handle distribution shifts, but typically optimize for a single error rate. Weighted conformal methods offer adaptation by reweighting calibration or past scores based on some notion of similarity to new observations (Lei & Wasserman, 2014; Guan, 2019; Tibshirani et al., 2019; Sesia & Romano, 2021; Han et al., 2022; Guan, 2023; Ghosh et al., 2023; Mao et al., 2024) improving local coverage. Bounds for non-exchangeable sequences (Barber et al., 2023) further suggest emphasizing calibration samples that are nearly exchangeable with the test point. This motivates our approach, which leverages weighted adaptive conformal quantiles to remain calibrated across time. Conformal prediction has also been applied to anomaly detection by thresholding arbitrary anomaly scores under exchangeability (Angelopoulos & Bates, 2021; Guan, 2019; Bates et al., 2023). However, existing methods do not simultaneously provide interpretable, distribution-agnostic anomaly scores, directly control alarm rates, and adapt robustly to non-exchangeable time series. Our work addresses this gap by developing a conformal anomaly detection framework that is both interpretable and resilient to real-world distribution shifts.

---

[2]informally, a sequence of observations is exchangeable if any permutation of the observations has the same joint probability

## 3 BACKGROUND

Consider $S \in \mathbb{R}$ a nonconformity score variable that quantifies the performance of a predictive model $h : \mathcal{X} \to \hat{\mathcal{Y}}$ on a joint distribution $P_{X,Y}$ using a nonconformity function $e : \mathcal{Y} \times \hat{\mathcal{Y}} \to \mathbb{R}$. The input $X \in \mathcal{X}$ represents the model's input space, $Y \in \mathcal{Y}$ denotes the true target variable, and $\hat{\mathcal{Y}}$ corresponds to the output space of the model, which may include predictions or derived statistics over $Y$. The nonconformity function $e$ measures the degree of disagreement between the true target and the model's predictions, enabling $S = e(Y, h(X))$ to capture how atypical a prediction is within the given distribution. An example of a nonconformity function for a point prediction model is absolute error $e(Y, \hat{Y}) = |Y - \hat{Y}|$.

### 3.1 CONFORMAL OUTLIER DETECTION.

In the context of anomaly detection we characterize the distribution of the non-conformity score variable $S \sim P_S$ where $S = e(Y, h(X)) \in \mathbb{R}$ under non-anomalous conditions $X, Y \sim P_{X,Y}$. [3] Observations are flagged as outliers (or anomalies) when the composition of the nonconformity function $e$ and the predictive model $h$ produces unusually high scores.[4] Given a significance level $\alpha$, which controls the tolerated false positive rate, an anomaly detection function $C_\alpha : \mathcal{X}, \mathcal{Y} \to \{0, 1\}$ should satisfy the following property:

$$\mathbb{P}(C_\alpha(X_{n+1}, Y_{n+1}) = 1) \leq \alpha \tag{1}$$

where $\mathbb{P}$ is the probability over unseen test data sampled from the non-anomalous distribution, $X_{n+1}, Y_{n+1} \sim P_{X,Y}$. In the standard split-conformal setting, we observe $\mathbf{s} = S_1, \ldots, S_n$ non-conformity scores derived from non-anomalous data, $S_i = e(Y_i, h(X_i))$ with $X_i, Y_i \sim P_{X,Y}$. Non-conformity scores need not be independent of each other; the following conformal anomaly detection function satisfies, under echangeability conditions[5], the false positive bound in equation 1:

$$C_\alpha(X_{n+1}, Y_{n+1}) = \mathbf{1}[S_{n+1} > \hat{q}_\alpha], \quad \hat{q}_\alpha = Q_{1-\alpha}(\sum_{i=1}^{n} \frac{1}{n+1}\delta_{S_i} + \frac{1}{n+1}\delta_\infty). \tag{2}$$

Here $\hat{q}_\alpha$ is the empirical conformal quantile, conservatively adjusted with a point mass at infinity.

**Conformal Outlier Detection Beyond Exchangeability** To account for heterogeneity in the non-conformity scores across the input space or potential temporal drift, we consider the generalized weighted conformal quantile estimate $\hat{q}_\alpha^w = \mathbb{Q}_{1-\alpha}(\mathbf{s}, \mathbf{w})$ defined as:

$$\mathbb{Q}_{1-\alpha}(\mathbf{s}, \mathbf{w}) = Q_{1-\alpha}(\sum_{i=1}^{n} \frac{w_i}{||\mathbf{w}||_1 + 1}\delta_{S_i} + \frac{1}{||\mathbf{w}||_1 + 1}\delta_\infty). \tag{3}$$

where $\mathbf{w} = \{w_i \in [0, 1]\}_{i=1}^{n}$ is a weighting vector applied to the calibration points. The standard result in Eq. 2 is recovered when $w_i = 1, \forall i = 1, \ldots, n$.

This weighted conformal quantile estimate produces a generalization of the conformal anomaly detector from equation 2. This conformal anomaly detection has false alarm rate guarantees even in non-exchangeable settings as described in the following proposition 3.1.

**Proposition 3.1.** *(Direct application of Theorem 2 and 3 in Barber et al. (2023) Given $\alpha \in (0, 1)$, $\mathbf{s} = \{S_i\}_{i=1}^{n+1}$ a set of non-conformity scores where $S_{n+1}$ corresponds to the test point, and a vector of weights $\mathbf{w} = \{w_i \in [0, 1]\}_{i=1}^{n}$ for the previous $n$ observations the detector*

$$A_{n+1} = C_{\alpha,\mathbf{w}}(X_{n+1}, Y_{n+1}) = \mathbf{1}[S_{n+1} > \hat{q}_\alpha^{\mathbf{w}}] \tag{4}$$

*based on the weighted conformal quantile estimate in Eq.3 satisfies the false alarm rate guarantees*

$$\begin{aligned} \mathbb{P}(A_{n+1} = 1) \quad &\leq \alpha + \sum_{i=1}^{n} \frac{w_i}{||\mathbf{w}||_1 + 1} d_{TV}(\mathbf{s}, \mathbf{s}^i) \\ &> \alpha - \sum_{i=1}^{n} \frac{w_i}{||\mathbf{w}||_1 + 1} d_{TV}(\mathbf{s}, \mathbf{s}^i) - \frac{1}{||\mathbf{w}||_1 + 1}. \end{aligned} \tag{5}$$

*Here $d_{TV}(\mathbf{s}, \mathbf{s}^i)$ is the distance in total variation between the sequence $\mathbf{s}$ ($n$ previously observed point and the test point $n + 1$) and $\mathbf{s}^i$ which denotes the sequence of non-conformity scores after swapping the test point $n + 1$ with the $i$-th previously observation. The lower bound is valid under the assumption that the non-conformity scores take equal values with probability 0.*

---

[3]Although $\mathcal{X}$ and $\mathcal{Y}$ are treated as separate spaces, they may overlap, as in reconstruction-error-based scores where $Y = X$.

[4]Unusually low scores can be handled similarly, nonconformity scores need not be positive

[5]The sequence $S_1, \ldots, S_{n+1}$ is exchangeable if $P(S_1, \ldots, S_{n+1}) = P(S_{\sigma(1)}, \ldots, S_{\sigma(n+1)})$ for any permutation $\sigma$

Intuitively, Proposition 3.1 indicates that one would like to assign higher weights to previous observations that are, pairwise, most exchangeable with the test sample (i.e., $P(S_1, \ldots, S_i, \ldots, S_{n+1}) \simeq P(S_1, \ldots, S_{n+1}, \ldots, S_i)$), and lower weights otherwise. Additionally, the lower bound encourages the maximization of $||\mathbf{w}||_1$ and therefore keeping the weights as close to one as possible. One could decide $\mathbf{w}$ if given access to prior knowledge about the values or reasonable upper bounds of $d_{TV}(\mathbf{s}, \mathbf{s}^i)$. In the context of time series, previous works such as Barber et al. (2023) have set $\mathbf{w}$ to exponentially decay with time ($w_i = \gamma^{n-i}$); in non-time-series settings, other works such as (Lei & Wasserman, 2014; Guan, 2019; Sesia & Romano, 2021; Han et al., 2022; Guan, 2023; Ghosh et al., 2023; Mao et al., 2024) decide the weights based on criteria such as distance in covariate space, or optimize them to guarantee a particular false positive rate coverage $\alpha$, (Han et al., 2022; Amoukou & Brunel, 2023). Next, we present our adaptive conformal score method, which learns $\mathbf{w}$ with the objective of providing scores that are calibrated for every feasible false alarm across time.

## 4 ADAPTIVE CONFORMAL ANOMALY SCORE

The conformal outlier detection framework provides a principled way to define a binary anomaly decision variable based on a preselected $\alpha$ with generalization guarantees. However, the underlying nonconformity score $S$ may not itself be an interpretable indicator of anomaly, particularly in sequential settings where its distribution may drift over time. To address this, we aim to learn an adaptive mapping that assigns each score an approximate probability of observing a more extreme value under prior (ideally normal) conditions, yielding a distribution-agnostic $p$-value estimate. Formally, we consider a time series setting with a sequence of nonconformity scores $S_1, \ldots, S_t$. In prediction-based anomaly detection, these are derived from a forecasting model $h : \mathbb{R}^{n_c \times n_f} \to \mathcal{Y}^d$, which maps a context of length $n_c$ with $n_f$ features to a $d$-step-ahead forecast $\hat{Y}_{t+1}^d = h_d(X_{t-n_c-d:t-d+1})$. The nonconformity score for sample $t+1$ at horizon $d$ is $S_{t+1}^d = |Y_{t+1} - \hat{Y}_{t+1}^d|$. For clarity, we omit the index $d$ in the following section, since the analysis applies independently to each prediction horizon, and reintroduce it later when needed.

### 4.1 CONFORMAL ANOMALY SCORE

We wish to learn a parametric mapping $\beta_{\mathbf{w}} : \mathbb{R}, \mathbb{R}^t \to [0, 1]$ of the previous nonconformity scores $\mathbf{s} = \{S_i\}_{i=1}^t$ and the test sample $S_{t+1}$; this mapping $\beta_{\mathbf{w}}$ should be such that it can be directly compared to any $\alpha$ threshold to produce an anomaly detector with the same false alarm rate guarantees as the one described in equations equation 1 and equation 2. Given a set of non-conformity scores derived from past, ideally non-anomalous data [6], their associated weights $\vec{w} = \{w_i \in [0, 1]\}_{i=1}^t$, and a non-conformity score test sample $S_{t+1}$ we propose the following score normalization

$$\beta_{\mathbf{w}}(S_{t+1}) = \sup\{\alpha \in [0, 1] : S_{t+1} \leq \mathbb{Q}_{1-\alpha}(\mathbf{s}, \mathbf{w})\}. \tag{6}$$

Here $\beta_{\mathbf{w}}(S_{t+1})$ can be interpreted as the weighted, conformalized p-value, $\beta_{\mathbf{w}}(S_{t+1}) = \beta_{t+1}$ (we omit the explicit dependence on $\mathbf{s}$ for clarity). The proposed function automatically maps an anomaly score $S$, which can take arbitrary real values, into a normalized score that directly relates to the desired false alarm rate. The decision of an anomaly detection threshold becomes interpretable for the end user (it directly translates into the desired false alarm level) and preserves the guarantees of the original conformal outlier detector as shown in Proposition 4.1.

**Proposition 4.1.** *Given $\alpha \in [0, 1]$, $\{S_i\}_{i=1}^{t+1}$ a set of exchangeable non-conformity scores, and their weights $\mathbf{w} = \{w_i =\in [0, 1]\}_{i=1}^t$ the detector $C_{\beta_{\mathbf{w}}}(X_{t+1}, Y_{t+1}) = \mathbf{1}[\beta_{\mathbf{w}}(S_{t+1}) < \alpha]$ based on the $\beta_{\mathbf{w}}(\cdot)$ mapping defined in equation 6 is equivalent to equation 4 and therefore satisfies the conformal false alarm rate guarantees presented in equation 5 in Proposition 3.1. Proof in Appendix B.*

### 4.2 ADAPTIVE WEIGHTED ANOMALY SCORES UNDER NON-EXCHANGEABILITY

Our proposed conformal anomaly score mapping $\beta_{\mathbf{w}}(\cdot)$ in equation 6 depends on the weights $\mathbf{w}$ assigned to the previously observed scores. Therefore, given a new observation $S_{t+1}$ the mapping can be directly expressed as a function of $\mathbf{w}$, $\beta_{\mathbf{w}}(S_{t+1}) = \beta_{t+1}(\mathbf{w})$ such that

$$\beta_{t+1}(\mathbf{w}) := \frac{1 + \sum_{k=j_{t+1}}^{n} w_{\pi^{-1}(k)}}{|\mathbf{w}| + 1}, \quad j_{t+1} = \sum_{i=1}^{t} \mathbf{1}[S_{t+1} \leq S_i]. \tag{7}$$

---

[6]For sequences containing a known fraction of anomalous samples below some upper bound $\alpha'$, the derivation follows similarly, but the interpretation of $\beta_{\mathbf{w}}(S_{t+1})$ is $\alpha + \alpha'$ where $\alpha$ is the lower bound of the p-value of the sample.

Where $\pi : [n] \rightarrow [n]$ represents a sorted mapping of the previous $n$ nonconformity scores such that $\pi(i) = k \in [n], \forall i \in [n]$ where $\pi(i) < \pi(j)$ if $S_i \leq S_j, \forall i \neq j$. $\pi^{-1}(k)$ is the inverse sorting operation, mapping $k$ to the index of the observation corresponding to the $k$ largest value.

We want our proposed conformal score to be well calibrated across time, meaning $\mathbb{P}(\beta_{\mathbf{w}}(S_{t+1}) \leq \alpha) \approx \alpha$, for all $\alpha \in [0,1]$ and $t$. In lieu of that, we require $\beta_{\mathbf{w}}(S_{t+1})$ to be a conservative estimate such that $\mathbb{P}(\beta_{\mathbf{w}}(S_{t+1}) \leq \alpha) \leq \alpha$. Such calibration ensures that the conformalized scores adapts effectively to distributional shifts over time. The ideal condition under non-anomalous distributions for $S_{t+1}$, $\mathbb{P}(\beta_{\mathbf{w}}(S_{t+1}) \leq \alpha) = \alpha, \forall \alpha \in [0,1]$ is achieved when $\beta_{\mathbf{w}}(S_{t+1}) \sim U_{[0,1]}$. We also note that $\beta_{\mathbf{w}}(S_{t+1})$ cannot produce non-trivial quantile estimates below its effective sample size $\alpha_c = \frac{1}{|\mathbf{w}|+1}$. We therefore seek to learn a set of feasible weights $\mathbf{w}$ satisfying these conditions by minimizing the 1-Wasserstein distance ($\mathcal{W}_1$) between the cumulative density function (CDF) of the proposed score variable $F_{\beta_{t+1}(\mathbf{w})}$, where $\beta_{t+1}(\mathbf{w}) = \beta_{\mathbf{w}}(S_{t+1})$ as in equation 7, and the CDF of the uniform distribution $F_U$, subject to an effective sample size constraint determined by our critical false alarm rate $\alpha_c$. Namely

$$\min_{\mathbf{w}} \mathcal{W}_1(F_{\beta_{t+1}(\mathbf{w})}, F_U) \quad s.t. \quad |\mathbf{w}| > \frac{1}{\alpha_c} - 1, w_i \in [0,1], \forall i \in [n]. \tag{8}$$

Here $\alpha_c$ is the user-defined critical false alarm rate. From the dual definition of $\mathcal{W}_1$ we have

$$\begin{aligned} \mathcal{W}_1(F_{\beta_{t+1}(\mathbf{w})}, F_U) &= \int_0^1 |F_{\beta_{t+1}(\mathbf{w})}^{-1}(p) - F_U^{-1}(p)|dp \\ &= \int_0^1 |F_{\beta_{t+1}(\mathbf{w})}(\alpha) - F_U(\alpha)|d\alpha \\ &= \mathbb{E}_{\alpha \sim U_{[0,1]}}|\mathbb{P}(\beta_{t+1}(\mathbf{w}) \leq \alpha) - \alpha|, \end{aligned} \tag{9}$$

which indicates that minimizing $\mathcal{W}_1(F_{\beta_{t+1}(\mathbf{w})}, F_U)$ is equivalent to minimizing the calibration gap $|\mathbb{P}(\beta_{t+1}(\mathbf{w}) \leq \alpha) - \alpha|$ uniformly across all false alarm rates. We next approximate the objective in equation 8 using finite samples and give the corresponding algorithm.

## 5 OPTIMIZATION

In practice, we need to approximate $F_{\beta_{t+1}(\mathbf{w})}(\alpha)$ in equation 8 with a finite number of samples $n_b$, which results in the following empirical CDF based on the scores $\{\beta_{t+j}\}_{j=1}^{n_b}$

$$\hat{F}_{\beta_{t+1}(\mathbf{w})}(\alpha) = \frac{1}{n_b} \sum_{j=1}^{n_b} \mathbf{1}[\beta_{t+j}(\mathbf{w}) \leq \alpha]. \tag{10}$$

Then, the $\mathcal{W}_1$ objective in equation 8 can be empirically approximated as follows

$$\mathcal{W}_1(\hat{F}_{\beta_{t+1}(\mathbf{w})}, F_U) = \sum_{k=1}^{n_b} \int_{\frac{k-1}{n_b}}^{\frac{k}{n_b}} |\beta_{t+\hat{\pi}^{-1}(k)}(\mathbf{w}) - \alpha|d\alpha, \tag{11}$$

where $\hat{\pi}$ is the sort mapping of $\{\beta_{t+j}(\mathbf{w})\}_{j=1}^{n_b}$ scores such that $\beta_{t+\hat{\pi}^{-1}(k)}(\mathbf{w}) \leq \beta_{t+\hat{\pi}^{-1}(k+1)}(\mathbf{w})$. Note that the expression in equation 11 is a sum of integrals of piecewise linear functions. Therefore, it is differentiable w.r.t. to each $\beta_{t+j}(\mathbf{w})$, and consequenlty w.r.t. to each $\mathbf{w}$ (see equation 4) and also computable in closed form. Then the weights can be updated using projected gradient descent

$$\begin{aligned} \mathbf{w}_{t+n_b+1} &= \mathbf{w}_{t+n_b} - \gamma \Big\{ \sum_{i=1}^{n_b} \frac{\partial \mathcal{W}_1}{\partial \beta_{t+i}} \frac{\partial \beta_{t+i}(\mathbf{w}_{t+n_b})}{\partial w_k} \Big\}_{k=1}^n \\ \mathbf{w}_{t+n_b+1} &= \prod_{\mathbf{w} \in [0,1]^n, |\mathbf{w}| > \frac{1}{\alpha_c} - 1} \Big[ \mathbf{w}_{t+n_b+1} \Big] \end{aligned} \tag{12}$$

Note that here $\mathbf{w}_t$ denotes our current estimate of the entire weighting vector $\mathbf{w}$ at time $t$. The partial derivatives can be expressed in closed form as

$$\frac{\partial \mathcal{W}_1}{\partial \beta_{t+i}} = \begin{cases} -\frac{1}{n_b}, & \text{if } \beta_{t+i} < \frac{\hat{\pi}(i)-1}{n_b}, \\ 2\beta_{t+i} - \frac{2\hat{\pi}(i)-1}{n_b}, & \text{if } \frac{\hat{\pi}(i)-1}{n_b} \leq \beta_{t+i} \leq \frac{\hat{\pi}(i)}{n_b}, \\ +\frac{1}{n_b}, & \text{if } \beta_{t+i} > \frac{\hat{\pi}(i)}{n_b}. \end{cases} \tag{13}$$

and

$$\frac{\partial \beta_{t+i}(\mathbf{w})}{\partial w_k} = \frac{-\beta_{t+i}(\mathbf{w}) + \mathbf{1}[j_{t+i} \leq \pi(k)]}{||\mathbf{w}||_1 + 1} \tag{14}$$

The derivatives themselves have a simple interpretation. The derivative of $\frac{\partial \mathcal{W}_1}{\partial \beta_{t+i}}$ pushes a normalized score $\beta_{t+i}$ to lie within the ranges of its empirical quantile bucket $[\frac{\hat{\pi}(i)-1}{n_b}, \frac{\hat{\pi}(i)}{n_b}]$, and is minimized when $\beta_{t+i} = \frac{2}{2n_b}\frac{\hat{\pi}(i)-1}{}$. The derivative $\frac{\partial \beta_{t+i}(\mathbf{w})}{\partial w_k}$ establishes that one can increase $\beta_{t+i}$ by decreasing the weight of scores higher than the currently-observed score $S_{t+1}$ or by globally decreasing the overall sample size $||\mathbf{w}||_1$.

---

**Algorithm 1** 1-Wasserstein Adaptive Conformal Anomaly Score

---

**Require:** $\{S_t\}_{t=1}^T$: Scores, $\alpha_c$: min false alarm rate, $n$ max past samples, $n_b$ min batch size

   **Output:** : $\boldsymbol{\beta} \in [0,1]^{T-n_c}$ normalized score vector

   $n_c = \frac{1}{\alpha_c} - 1$, $\mathbf{w} = \{w_i = \mathbf{1}[i \leq n_c]\}_{i=1}^n$. # Compute critical samples and init weights

   $\mathbf{J}\boldsymbol{\beta}(\mathbf{w}) \leftarrow \{0\}^{n_b \times n}$, $i_b = 0$, $\boldsymbol{\beta} \leftarrow \{\}$ # Initialize score Jacobian, batch counter and output

   **for** $t = n_c : T - n_c$ **do**

     $\mathbf{s} = \{S_i\}_{i=\max(t-n,1)}^t$, $\hat{\mathbf{w}} = \{\hat{w}_i = w_{|\mathbf{s}|+1-i}\}_{i=1}^{|\mathbf{s}|}$ # Get past scores and corresponding weights

     $\pi \leftarrow \text{ARGSORT}(\mathbf{s})$ # sort past scores in ascending order

     $j_{t+1} = \sum_{s \in \mathbf{s}} \mathbf{1}[S_{t+1} < s]$, $\beta_{t+1} = \frac{\sum_{k=j_{t+1}}^{|\mathbf{s}|} \hat{w}_{\pi^{-1}(k)}+1}{||\hat{\mathbf{w}}||_1+1}$ # Compute p-value score for $S_{t+1}$

     $\boldsymbol{\beta} \leftarrow \boldsymbol{\beta} \cup \beta_{t+1}$, $i_b \leftarrow i_b + 1$

     $\mathbf{J}\boldsymbol{\beta}(\mathbf{w})_{i_b,n-k} = \{\frac{\partial \beta_{t+1}}{\partial \hat{w}_k}\}$ for $k = 1, ..., |s|$, using equation 14 # Compute partial derivatives

     **if** $i_b = n_b$ **then**

       $\hat{\pi} \leftarrow \text{ARGSORT}(\boldsymbol{\beta}_{t+1-n_b:t+1})$ #Sort last $n_b$ normalized scores and compute gradient

       Compute $\{\frac{\partial \mathcal{W}_1}{\partial \beta_{t+i}}\}_{i=1}^{n_b}$ using $\hat{\pi}$, equation 13, $\nabla \mathcal{W}_1(\mathbf{w}) = \{\sum_{i=1}^{n_b} \frac{\partial \mathcal{W}_1}{\partial \beta_{t+i}} \mathbf{J}\boldsymbol{\beta}(\mathbf{w})_{i,k}\}_{k=1}^n$

       $\mathbf{w} \leftarrow \prod_{\mathbf{w} \in [0,1]^n, |\mathbf{w}| > n_c} \left[ \mathbf{w} - \gamma \nabla \mathcal{W}_1(\mathbf{w}) \right]$, $i_b \leftarrow 0$

     **end if**

   **end for**

---

We propose $\mathcal{W}_1$-ACAS (Algorithm 1), which operates by sequentially estimating normalized scores $\beta_t$ using the current weight estimates. The weights $\mathbf{w}$ are then periodically updated to minimize the objective in Eq. 8, based on the online sample buffer and the update rules in Eqs. 12, 13 and 14.

**Aggregation Across Multiple Forecast Horizons** We extend Algorithm 1 to operate across multiple forecast horizons. Specifically, we run $D$ parallel instances of the algorithm, each associated with a $d$-step ahead prediction error, $S_{t+1}^d = |Y_{t+1} - \hat{Y}_{t+1}^d|$, with $\hat{Y}_{t+1}^d = h_d(Y_{t-n_c-d:t-d+1})$, $d \in [D]$. This produces a set of $D$ conformal p-values for each observation $t + 1$, denoted $\{\beta_{t+1}^d\}_{d \in [D]}$. The final anomaly score is the median across horizons,

$$\bar{\beta}_{t+1} = \text{median}_{d \in [D]} \beta_{t+1}^d, \qquad \beta_{t+1}^d = \beta_{\mathbf{w}^d}(S_{t+1}^d). \tag{15}$$

This requires an observation to be identified as a significant outlier by more than half of the horizon-specific detectors. In the streaming setting, we maintain a buffer of forecasts at different horizons. When a new sample $Y_{t+1}$ is observed, we collect its aligned forecasts $\{\hat{Y}_{t+1}^d\}_{d \in [D]}$, compute the corresponding errors $\{S_{t+1}^d\}_{d \in [D]}$, and update each horizon-specific instance of Algorithm 1 to obtain the adaptive p-values, $\{\beta_{t+1}^d\}_{d \in [D]}$. In Appendix C.2.4 we describe how Algorithm 1 extends to multivariate time series anomaly detection in a similar manner. We also outline several standard p-value combination techniques, which can also be applied to aggregate the horizon-specific p-values.

## 6 EXPERIMENTS

We evaluate the proposed conformalized anomaly score $\mathcal{W}_1$-ACAS (Algorithm 1) by analyzing its calibration and anomaly detection performance on time series data. Synthetic experiments (Appendix C.1) validate its ability to remain calibrated under both gradual and abrupt distribution shifts, where ground-truth p-values are available. Our main empirical study focuses on real-world anomaly detection datasets, where we assess detection accuracy using both threshold-independent and threshold-dependent metrics.

**Anomaly Detection Datasets.** We evaluated the performance of our proposed method ($\mathcal{W}_1$-ACAS, Algorithm 1) for unsupervised univariate time series anomaly detection when applied to a

pre-trained time series foundation model. Experiments are conducted on seven benchmark datasets: YAHOO (Laptev et al., 2015), NEK (Si et al., 2024), NAB (Ahmad et al., 2017), MSL (Lai et al., 2021), IOPS (IOPS, n.d.), STOCK (Tran et al., 2016), and WSD (Zhang et al., 2022), all part of the curated anomaly detection benchmark of Liu & Paparrizos (2024). For the multivariate experiments, we additionally use the curated subsets of TAO (Laboratory, 2024), GECCO (Rehbach et al., 2018), LTDB (Goldberger et al., 2000), and Genesis (von Birgelen & Niggemann, 2018) released as part of the benchmark in Liu & Paparrizos (2024). Each dataset consists of an initial segment without anomalies used for training or calibration, followed by a test split that may contain anomalies.

$\mathcal{W}_1$-**ACAS + TSFM.**   We integrate $\mathcal{W}_1$-ACAS with three pre-trained TSFMs: Tiny Time Mixers (TTM) (Ekambaram et al., 2024), Chronos-Bolt-Small (Chronos) (Ansari et al., 2024), and TiRex (Auer et al., 2025). All models use a context length of 52 and a forecast horizon of $D = 15$. For Algorithm 1, we set the critical false alarm rate to $\alpha_c = 0.01$, batch size $n_b = 10$, and learning rate $\gamma = 0.001$. We use ADAM (Kingma & Ba, 2015) to perform an adaptive gradient descent on the weights $\mathbf{w}$. Appendix C.2.3, Fig. 8, analyzes the impact of aggregating forecast horizons, showing that $D = 15$ provides a reasonable balance between performance and sample efficiency. Figures 9, 10, and 11 show the sensitivity of $\mathcal{W}_1$-ACAS to the learning rate $\gamma$, batch size $n_b$, and $\alpha_c$. The method shows low variability for small $\gamma$ and $n_b$. The parameter $\alpha_c$ controls the maximum acceptable $p$-value resolution: smaller values require a larger number of in-distribution past observations $n_c$, but do not impose a lower bound on the detectable anomaly level.

**Baseline Methods.**   We compare $\mathcal{W}_1$-ACAS against two TSFM-based baselines: a **Gaussian** model that fits the mean absolute forecast error across $d$ steps using calibration data, and a **Conformal** offline approach that learns $p$-value mappings per horizon and aggregates them by the median. We also include top-performing classical methods from Liu & Paparrizos (2024): **KShape** (Paparrizos & Gravano, 2015; 2017; Boniol et al., 2021), **POLY** (Li et al., 2007), **Sub-PCA** (Aggarwal & Aggarwal, 2017), **Sub-KNN** (Ramaswamy et al., 2000), and **SAND** (Boniol et al., 2021). We further include strong semi-supervised deep learning–based anomaly detection methods (Audibert et al., 2022), namely **CNN** (Munir et al., 2018), **USAD** (Audibert et al., 2020), and **OmniAnomaly** (Su et al., 2019), as well as the recent general purpose TSFM **MOMENT** (Goswami et al., 2024), which provides zero-shot anomaly scoring. Additional details are provided in Appendix C.2.1.

**Evaluation Metrics.**   We report both point-wise (AUC, PA-F1) (Wu et al., 2022; Wang et al., 2024; Liu & Paparrizos, 2024) and range-wise metrics (VUS (Paparrizos et al., 2022a), Affiliation-F1 (Huet et al., 2022)). For threshold-dependent scores (PA-F1, Affiliation-F1), we follow the oracle strategy of Liu & Paparrizos (2024), selecting the best threshold in $[0, 1]$ and reporting the associated False Positive Rate (FPR) and calibration error (CalErr). Further details are in Appendix C.2.2.

**Results**   Table 1 reports the average performance of $\mathcal{W}_1$-ACAS, applied to different TSFM models, compared against the described baselines  on the univariate datasets. Our method achieves the strongest performance on threshold-dependent metrics (PA-F1, Affiliation-F), including when compared with semi-supervised methods such as CNN, USAD, and OmniAnomaly, while remaining competitive on threshold-independent metrics (AUC, VUS). When conditioned on the same TSFM model, $\mathcal{W}_1$-ACAS shows clear improvements over the Gaussian and Conformal baselines.   Figure 2 shows the average performance per univariate dataset for a subset of the methods, extended per-dataset results are provided in Tables 2, 3 and 4 in Appendix C.2.3. Table 5 shows that TSFM models have similar prediction errors across datasets, consistent with their comparable anomaly detection performance. Results for the multivariate datasets are presented in Table 6 in Appendix C.2.4, where we demonstrate how our approach naturally extends to the multivariate setting via $p$-value aggregation, achieving top performance relative to the corresponding baselines.

Figure 3 shows the FPR–threshold curves in the low-FPR regime, where $\mathcal{W}_1$-ACAS (blue) yields the most conservative thresholds, staying closer to or below the identity line compared to competing methods, while also exhibiting the lowest variance. Figure 4 shows representative detection examples along with the final learned weights. We observe that $\mathcal{W}_1$-ACAS is adapted to capture underlying temporal patterns in errors if present. Moreover, our method effectively identifies a transition in score distributions (e.g., in the vicinity of an anomalous region) but then quickly adapts to the new anomalous distribution; this helps minimize the number of alarms in the end-to-end system.

Additional examples are provided in Appendix C.2.3: Figure 6 shows more detection cases, and Figure 7 illustrates the trade-offs between FPR and F1 scores (PA-F1, Affiliation-F) across datasets. The operating points of $\mathcal{W}_1$-ACAS (blue), in most cases, achieve both the highest F1 score and

Table 1: **Performance Summary across univariate datasets.** Entries indicate the mean $\pm$ standard deviation computed by first averaging within each dataset group, then averaging across groups (equal weight). Higher numbers are better for PA-F1, Affiliation-F, AUC-PR, VUS-PR; lower numbers are better for FPR, and calibration error (CalErr). Underlined results indicate best post-hoc methods applied to the same base forecaster, while bold indicate best results overall. Methods marked with * denote deep learning semi-supervised approaches.

| Forecaster | AD Method | PA-F1 ↑ | Affiliation-F ↑ | FPR ↓ | CalErr ↓ | AUC-PR ↑ | VUC-PR ↑ |
|---|---|---|---|---|---|---|---|
| Chronos | $\mathcal{W}_1$-ACAS | 0.912 ± 0.066 | 0.893 ± 0.060 | **0.077 ± 0.114** | **0.025 ± 0.029** | **0.355 ± 0.261** | 0.440 ± 0.272 |
| Chronos | Conformal | 0.863 ± 0.109 | 0.891 ± 0.063 | 0.111 ± 0.130 | 0.038 ± 0.055 | 0.310 ± 0.240 | 0.420 ± 0.248 |
| Chronos | Gaussian | 0.716 ± 0.260 | 0.842 ± 0.066 | 0.123 ± 0.109 | 0.075 ± 0.061 | 0.265 ± 0.250 | 0.438 ± 0.245 |
| TTM | $\mathcal{W}_1$-ACAS | 0.889 ± 0.108 | 0.886 ± 0.058 | 0.082 ± 0.120 | 0.029 ± 0.031 | 0.342 ± 0.261 | 0.449 ± 0.245 |
| TTM | Conformal | 0.851 ± 0.124 | 0.885 ± 0.062 | 0.120 ± 0.145 | 0.044 ± 0.056 | 0.317 ± 0.247 | 0.448 ± 0.250 |
| TTM | Gaussian | 0.733 ± 0.240 | 0.849 ± 0.067 | 0.128 ± 0.115 | 0.081 ± 0.065 | 0.270 ± 0.261 | 0.450 ± 0.249 |
| TiRex | $\mathcal{W}_1$-ACAS | **0.925 ± 0.048** | **0.897 ± 0.064** | 0.084 ± 0.113 | 0.025 ± 0.031 | 0.344 ± 0.269 | 0.438 ± 0.272 |
| TiRex | Conformal | 0.878 ± 0.085 | 0.890 ± 0.063 | 0.107 ± 0.137 | 0.038 ± 0.055 | 0.308 ± 0.257 | 0.429 ± 0.256 |
| TiRex | Gaussian | 0.714 ± 0.264 | 0.837 ± 0.068 | 0.119 ± 0.103 | 0.090 ± 0.071 | 0.270 ± 0.264 | 0.432 ± 0.250 |
| - | POLY | 0.527 ± 0.276 | 0.848 ± 0.072 | 0.334 ± 0.269 | 0.282 ± 0.130 | 0.044 ± 0.031 | 0.377 ± 0.207 |
| - | Sub-KNN | 0.479 ± 0.291 | 0.786 ± 0.074 | 0.451 ± 0.276 | 0.174 ± 0.124 | 0.118 ± 0.106 | 0.321 ± 0.234 |
| - | KShape | 0.533 ± 0.299 | 0.789 ± 0.096 | 0.508 ± 0.291 | 0.176 ± 0.132 | 0.125 ± 0.135 | 0.303 ± 0.262 |
| - | PCA | 0.536 ± 0.332 | 0.826 ± 0.097 | 0.374 ± 0.297 | 0.248 ± 0.131 | 0.100 ± 0.093 | 0.417 ± 0.274 |
| - | SAND | 0.460 ± 0.309 | 0.790 ± 0.079 | 0.511 ± 0.296 | 0.134 ± 0.048 | 0.101 ± 0.117 | 0.289 ± 0.190 |
| - | CNN | 0.858 ± 0.138 | 0.881 ± 0.059 | 0.083 ± 0.103 | 0.643 ± 0.227 | 0.269 ± 0.292 | 0.423 ± 0.289 |
| - | OmniAnomaly | 0.674 ± 0.282 | 0.855 ± 0.068 | 0.209 ± 0.171 | 0.571 ± 0.187 | 0.166 ± 0.087 | 0.429 ± 0.317 |
| - | USAD | 0.498 ± 0.333 | 0.809 ± 0.099 | 0.425 ± 0.298 | 0.324 ± 0.161 | 0.088 ± 0.088 | 0.398 ± 0.262 |
| - | MOMENT_ZS | 0.596 ± 0.305 | 0.867 ± 0.088 | 0.261 ± 0.292 | 0.417 ± 0.198 | 0.110 ± 0.075 | **0.461 ± 0.162** |

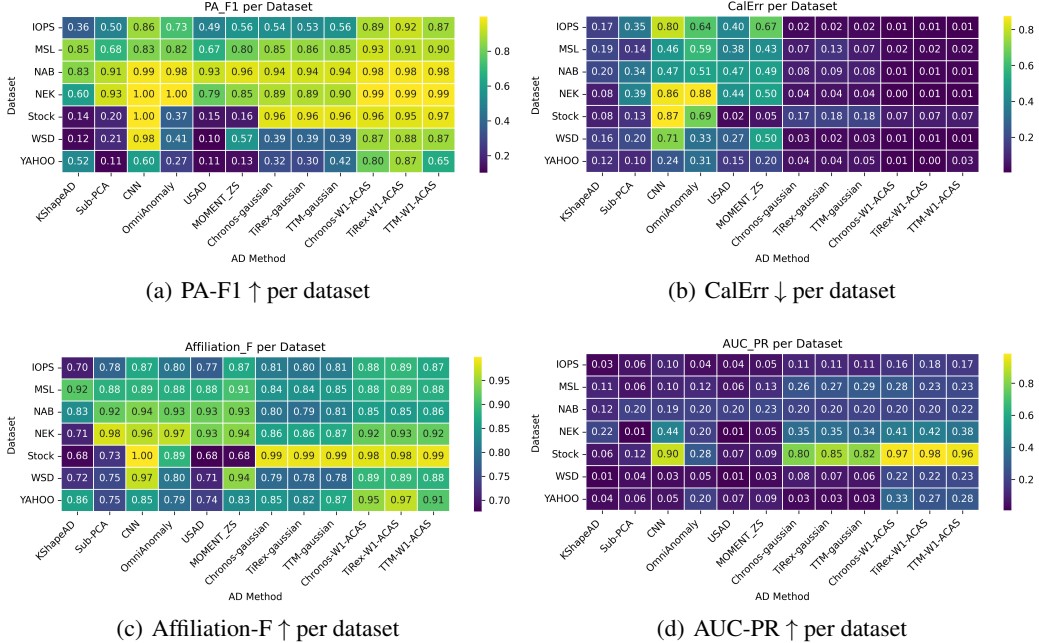

(a) PA-F1 ↑ per dataset

(b) CalErr ↓ per dataset

(c) Affiliation-F ↑ per dataset

(d) AUC-PR ↑ per dataset

Figure 2: **Performance across univariate datasets for a subset of anomaly detection methods.** Heatmaps show the average per-dataset performance for PA-F1, Affiliation-F, AUC-PR, and Calibration Error (CalErr) across a selected subset of methods. Higher values indicate better performance for PA-F1, Affiliation-F, and AUC-PR, while lower values are preferred for CalErr. Overall, the proposed $\mathcal{W}_1$-ACAS combined with Chronos, TiRex or TTM yields consistently low calibration error while remaining among the top-performing approaches. Note that CNN, OmniAnomaly and USAD are semi-supervised methods trained on the non-anomalous training datasplit.

lowest FPR, especially for PA-F1. Within each TSFM model, our method dominates its Gaussian (green) and Conformal (orange) counterparts in nearly all cases. Furthermore, it produces better-calibrated scores (low CalErr), making threshold selection more reliable in practice.

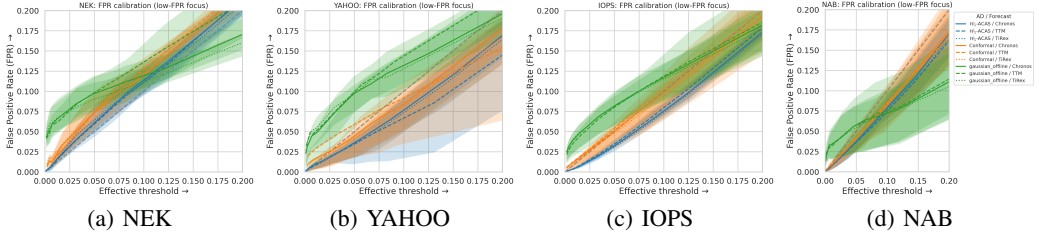

Figure 3: **FPR vs. threshold in the low-FPR regime.** Curves shows the mean false positive rate (FPR) across datasets for a given method, with shaded inter-quartile range (IQR) bands. The dashed gray line indicates ideal calibration ($FPR = \beta$). Curves above the line reflect over-confident scoring (FPR larger than threshold), while curves below the line reflect conservative scoring. In most cases, $\mathcal{W}_1$-ACAS (blue) yields the most conservative thresholds, staying closer to or below the identity line compared to competing methods, while also having the lowest variance.

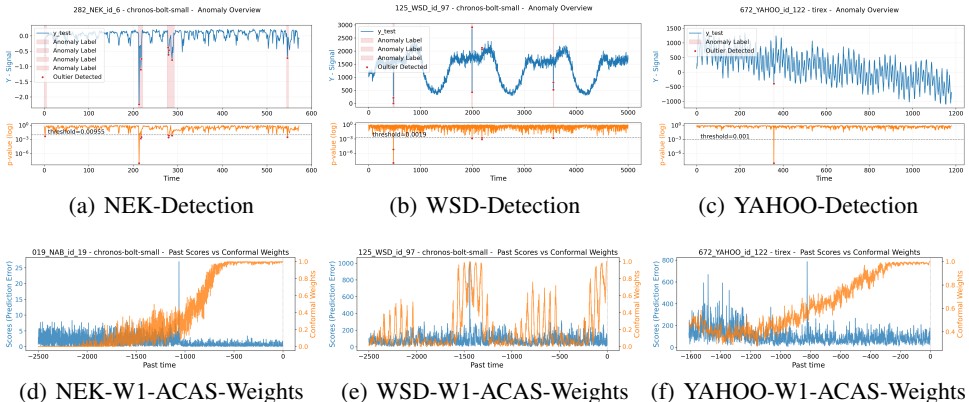

Figure 4: Example signals (blue) with ground-truth anomaly labels (red shading) are shown in the first row, where detected outliers (red dots) occur when adaptive $p$-values (orange) fall below a threshold under our proposed $\mathcal{W}_1$-ACAS method. The second row shows the final adaptive weights (orange) over past errors (blue), averaged across horizons, illustrating how $\mathcal{W}_1$-ACAS adapts to and captures underlying error patterns

## 7 CONCLUSION

In this paper, we presented $\mathcal{W}_1$-ACAS, a post-hoc adaptive conformal anomaly detection framework that leverages predictions from pretrained TSFMs to provide interpretable, distribution-agnostic, and well-calibrated anomaly scores without requiring retraining or large datasets. Experiments on benchmark datasets show that our method consistently outperforms competing baselines. $\mathcal{W}_1$-ACAS yields more conservative and stable thresholds, its a principled and easily applicable approach that adapts online to temporal error patterns, and minimizes false alarms by adjusting to distributions shifts. These properties make it especially suited for online monitoring in industrial and data-scarce environments. Future work will explore refining conformal weighting with contextual features, with straightforward extensions to multivariate anomalies via horizon-style aggregation.

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
