## A  RELATED WORK EXTENDED

**Time Series Anomaly Detection**    A key class of anomaly detection methods is prediction-based Giannoni et al. (2018), where anomalies are identified by deviations between predicted and observed values. These approaches assume that a well-trained forecaster captures normal temporal patterns, and significant prediction errors indicate potential anomalies Boniol et al. (2024). Such methods can in principle capture both point anomalies, where individual values deviate sharply, and contextual anomalies, where deviations only emerge relative to surrounding context Boniol et al. (2024). Given our focus on unsupervised settings with limited historical data, we build on pretrained forecasting models. Recent Time Series Foundation Models (TSFMs), trained at scale for forecasting, are particularly well suited for online detection in data-scarce scenarios Rasul et al. (2023; 2024); Ansari et al. (2024); Liang et al. (2024). In this work, we leverage three representative TSFMs: Tiny Time Mixers (TTM) (Ekambaram et al., 2024), based on the TSMixer architecture; Chronos-Bolt-Small (Chronos) (Ansari et al., 2024), a transformer-based model; and TiRex (Auer et al., 2025), which leverages an xLSTM architecture.

Recent benchmarks have evaluated the effectiveness of time series anomaly detection methods. The study by Liu & Paparrizos (2024) found that in unsupervised settings, classical distance-based and density-based approaches Li et al. (2007); Ramaswamy et al. (2000); Aggarwal & Aggarwal (2017); Paparrizos & Gravano (2015; 2017); Boniol et al. (2021) often outperform more complex models. However, these methods typically require access to the entire dataset (i.e., anomaly detections are non-causal and occur after the fact) and are not inherently designed for streaming applications Boniol et al. (2024). They may also struggle to capture richer temporal structures in the data, which limits their effectiveness in dynamic environments. Another critical challenge concerns the interpretability of anomaly scores and the choice of thresholds. Many evaluation studies emphasize threshold-independent metrics Schmidl et al. (2022); Paparrizos et al. (2022b); Goswami et al. (2022), yet the scores themselves often lack clear probabilistic meaning. Common thresholding strategies, such as standard deviation-based rules, depend on statistics computed over the entire dataset, making them impractical for streaming scenarios Ahmad et al. (2017).

In real-world deployments, an anomaly detection system must not only detect anomalies but also provide interpretable confidence scores while minimizing false alarms Cook et al. (2019). A high false alarm rate can overwhelm monitoring systems, reducing their practical utility. Our work addresses these challenges by developing an approach that enables adaptive thresholding in streaming environments while ensuring reliable anomaly detection, regardless of whether the anomalies are point-based or contextual.

**Conformal Prediction**    Conformal prediction methods Vovk et al. (2005) have gained significant attention for their ability to provide distribution-free uncertainty quantification with finite-sample generalization guarantees Shafer & Vovk (2008); Angelopoulos & Bates (2021). Among these, split conformal prediction (SCP) Papadopoulos et al. (2002) is a particularly appealing post-hoc, model-agnostic technique that requires only the model's predictions and a calibration dataset. SCP estimates an empirical quantile of a nonconformity score measuring how well the model's predictions align with the data to construct prediction sets that achieve the desired coverage. However, these guarantees rely on the exchangeability assumption [7] between calibration and test observations, which often does not hold in time series settings.

For non-exchangeable data, particularly time series, several adaptive conformal prediction methods have been proposed Gibbs & Candes (2021); Zaffran et al. (2022); Gibbs & Candès (2024). These approaches dynamically adjust the estimated quantile to correct for distribution shifts and achieve the target coverage level. However, they are typically designed for a single error rate objective, often optimizing the pinball loss or a surrogate function. In contrast, our work focuses on an adaptive method that remains effective across all error rates and desired alarm rate.

Weighted conformal quantile estimation Gibbs et al. (2023), where the calibration or past nonconformity scores are weighted differently has been used to achieve local coverage when the distribution of the error differs across the input space. Essentially, for any given observation, scores of samples that are similar to that observation get up-weighted, usually based on some metric (e.g., proximity in the covariate space) (Lei & Wasserman, 2014; Guan, 2019; Tibshirani et al., 2019; Sesia & Romano, 2021; Han et al., 2022; Guan, 2023; Ghosh et al., 2023; Mao et al., 2024), weights

---

[7]informally, a sequence of observations is exchangeable if any permutation of the observations has the same joint probability

can also be optimized to capture the variance of the non-conformity score across the input space (Han et al., 2022; Amoukou & Brunel, 2023). In the context of non-exchangeable data, Barber et al. (2023) derived a coverage bound linking the weights associated with a calibration sample and the total variation distance between the observed sequence and one where the calibration sample is swapped with the test sample. This bound suggests one should up-weight samples that are 'nearly exchangeable' with the new observation on a pairwise basis. This is the main inspiration for our proposed approach.

Conformal prediction has been explored for anomaly detection by setting thresholds on arbitrary anomaly scores from non-anomalous data while assuming exchangeability Angelopoulos & Bates (2021); Guan (2019); Bates et al. (2023). However, to the best of our knowledge, there is no existing method that simultaneously (i) seamlessly applies these techniques to generate interpretable, distribution-agnostic anomaly scores, (ii) directly translates scores into a desired alarm rate, and (iii) is inherently adapted to operate under non-exchangeability assumptions. Our work aims to bridge this gap by developing a conformal anomaly detection framework that is both interpretable and robust to real-world time series shifts.

## B    PROOFS

**Proof Proposition 4.1**    We show the equivalence of the detector $C_{\beta_\mathbf{w}}$ and the conformal outlier detector in equation 2 over the non-conformity scores $S$ by proving the following

$$
\begin{aligned}
C_{\beta_\mathbf{w}}(X_{t+1}, Y_{t+1}) \quad &= \mathbf{1}[\beta_\mathbf{w}(S_{t+1}) > 1 - \alpha] \\
&= \mathbf{1}[S_{t+1} > \mathbb{Q}_{1-\alpha}(\mathbf{s}, \mathbf{w})]
\end{aligned}
\tag{16}
$$

which involves proving that the events $\beta_\mathbf{w}(S_{t+1}) < \alpha$ and $S_{t+1} > \mathbb{Q}_{1-\alpha}(\mathbf{s}, \mathbf{w})$ are equivalent.

If $\beta_\mathbf{w}(S_{t+1}) = \beta_{t+1} < \alpha$ then $S_{t+1} > \mathbb{Q}_{1-\alpha}(\mathbf{s}, \mathbf{w})$ since by definition of $\beta_\mathbf{w}(\cdot)$ in equation 6 then $\beta_{t+1}$ is the maximum value in [0,1] that satisfies the quantile upper bound.

If $\beta_\mathbf{w}(S_{t+1}) = \beta_{t+1} \geq \alpha$ and since $S_{n+1} \leq \mathbb{Q}_{1-\beta_{t+1}}(\mathbf{s}, \mathbf{w}) \leq \mathbb{Q}_{1-\alpha'}(\mathbf{s}, \mathbf{w}), \forall \alpha' \leq \beta_{t+1}$ we have that $S_{t+1} \leq \mathbb{Q}_{1-\alpha}(\mathbf{s}, \mathbf{w})$.

## C    ADDITIONAL EXPERIMENTS

### C.1    SIMULATED EXAMPLES

We consider a similar simulated setting as Gibbs et al. (2023) to empirically evaluate the performance of the proposed method across time. We analyze a simple scenario where we observe a sequence of random variables $\{Y_t\}_{t=1}^T$, where $Y_t \sim \mathcal{N}(\mu_t, 1)$. We assume that our predictive model $h$ outputs a constant $\hat{Y}_t = 0, \forall t$. Then the error is $\epsilon_t = Y_t - \hat{Y}_t \sim \mathcal{N}(\mu_t, 1)$ and its distribution changes across time based on $\mu_t$. The nonconformity score is $s_t = |\epsilon_t|, \forall t$. We consider two different settings for the sequence of means $\{\mu_t\}_{t=1}^T$:

- **Random shift setting:** $\mu_t$ drifts continuously across time. Specifically, we set $\mu_0 = 0$ and

$$
\mu_{t+1} = \mu_t + \frac{1}{2}(\mu_t - \mu_{t-1}) + \frac{1}{2}\epsilon_t, \{\epsilon_t\} \sim \mathcal{N}(0, 0.05), \forall t.
\tag{17}
$$

- **Jump shift setting:** $\mu_t$ undergoes abrupt discontinuities every 500 time steps where $\mu_t$ increases by one step 15 times, and then starts decreasing by 1,

$$
\mu_t = \lfloor t/500 \rfloor \mathbf{1}[\lfloor t/500 \rfloor < 15] + [15 - \lfloor t/500 \rfloor] \mathbf{1}[\lfloor t/500 \rfloor \geq 15].
\tag{18}
$$

Given an observed non-conformity score $s_t = |\epsilon_t|$ we can compute its corresponding p-value $\alpha_t$ such that $\mathbb{P}_{\epsilon_t \sim \mathcal{N}(\mu_t, \infty)}(|\epsilon_t| > s_t) = \alpha_t = 1 - \Phi(s_t - \mu_t) + \Phi(-s_t - \mu_t)$ and compare it with the one estimated by the proposed normalized anomaly score $\beta_\mathbf{w}(s_t)$.

Figures 5.a and 5.b illustrate a sample of the generated signals under the Random Shift and Jump Shift settings. Each sequence consists of $T = 6000$ time steps, and our results are averaged over 15 independent realizations. We assess the performance of our proposed approach, Algorithm 1, referred to as $\mathcal{W}_1$-ACAS, with parameters $n = 2000$, $\alpha_c = 0.01$, and $n_b = n_c = \lceil \frac{1}{\alpha_c} - 1 \rceil$. We compare it against two baseline methods: (i) an adaptive conformal approach that assigns equal

weights of 1 to the most recent 2000 samples (ACAS with a fixed window) and (ii) a naive split conformal approach that computes scores using only the initial 100 samples (Split Conformal with fixed calibration).

In Figures 5.c and 5.d, we compare the empirical CDFs of each method against the empirical CDF of the ground truth p-values (denoted as Ground Truth), which naturally aligns with the identity line (reference). Notably, $\mathcal{W}_1$-ACAS demonstrates superior calibration, consistently aligning closely with the ground truth CDF and outperforming the other approaches.

Figures 5.e and 5.f present the average error of the scores of each method with respect to the ground truth p-values, across different bucket ranges of size 0.1 within $[0, 1]$. Specifically, we evaluate $\mathbb{E}[|\beta_{\mathbf{w}}(s_{t+1}) - \alpha_{t+1}| \mid \alpha_t \in [\alpha_l, \alpha_u]]$, where $\alpha_{t+1}$ represents the ground truth p-value for observation $t + 1$. The results indicate that $\mathcal{W}_1$-ACAS consistently outperforms the baseline methods, highlighting the advantages of an adaptive approach that dynamically learns how to weight past observations in a principled manner, rather than relying on a fixed number of past samples.

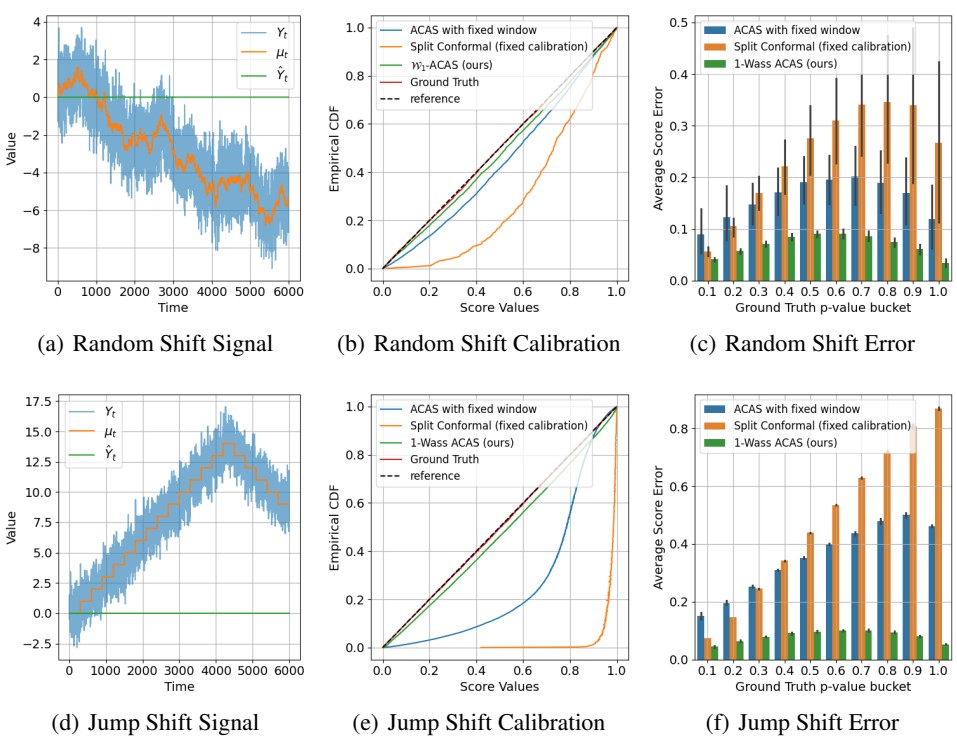

Figure 5: Figures (a) and (d) show an example of a generated signal under the random shift and jump shift settings with a sequence length of $T = 6000$. $\mu_t$ is the expected value of the observed signal, $Y_t$ is the observed signal $Y_t \sim \mathcal{N}(\mu_t, 1)$, and $\hat{Y}_t = 0$ the predicted value of a naive constant forecaster. Figures (b) and (e) show the empirical cumulative distribution functions (CDFs) of the various calibration approaches compared to the ground truth p-values (Ground Truth), which aligns with the idealized uniform CDF (reference). Results are averaged over 15 realizations. $\mathcal{W}_1$-ACAS demonstrates superior calibration, closely matching the ground truth distribution and improving upon the reference split conformal method (computed over calibration samples) and a fixed window ACAS method. Figures (c) and (f) show the average absolute error of the scores of the different methods with respect to the ground truth p-values, evaluated across bucket ranges of size 0.1 in $[0, 1]$. $\mathcal{W}_1$-ACAS consistently achieves lower estimation errors, highlighting the effectiveness of its adaptive weighting strategy with minimum parameters.

## C.2 ANOMALY DETECTION REAL DATASETS

### C.2.1 BASELINE METHODS

We compare $\mathcal{W}_1$-ACAS against two TSFM-based baselines. The first fits a **Gaussian** distribution to the mean absolute forecast error across $d$ steps using the calibration portion and assigns anomaly scores via the resulting $p$-values. The second applies a **Conformal** offline approach that learns a $p$-value mapping from the calibration split for each $d$ and aggregates the scores by the median. These baselines provide simple references built directly on TSFM errors.

We additionally consider several classic anomaly detection methods reported as top-performing in Liu & Paparrizos (2024):

- **KShape** (Paparrizos & Gravano, 2015; 2017; Boniol et al., 2021), which clusters subsequences via the k-Shape algorithm and scores anomalies by their distance to cluster centroids;
- **POLY** (Li et al., 2007), which fits a polynomial to the series and applies a GARCH model to residuals to estimate volatility;
- **Sub-PCA** (Aggarwal & Aggarwal, 2017), which projects subsequences onto a lower-dimensional hyperplane and scores deviations;
- **Sub-KNN** (Ramaswamy et al., 2000), which scores each instance by its distance to the $k$-th nearest neighbor;
- **SAND** (Boniol et al., 2021), an online method that adaptively down-weights older subsequences.
- **CNN** (Munir et al., 2018), is a causal convolutional forecasting model, the anomaly score is the prediction error. It is trained on non-anomalous data.
- **USAD** (Audibert et al., 2020) is an adversarially trained dual–autoencoder model learned on non-anomalous data, where anomaly scores are computed from a weighted reconstruction loss.
- **OmniAnomaly** (Su et al., 2019) is a stochastic recurrent VAE that incorporates GRU dynamics, planar normalizing flows, and temporal latent stochasticity; anomaly scores are derived from reconstruction probabilities. It is trained on non-anomalous data.
- **MOMENT** (Goswami et al., 2024) is a general-purpose TSFM based on a T5-style encoder trained via masked time-series modeling. It supports zero-shot anomaly scoring using masked-token reconstruction error and is pretrained on a broad corpus including anomaly detection datasets (Liu & Paparrizos, 2024).

For these approaches we adopt the implementations from Liu & Paparrizos (2024) with the best reported hyperparameters and their default $[0, 1]$ min–max normalization fitted on the full dataset. For consistency with our $p$-value scoring (where lower values indicate greater anomaly), we take one minus the reported score. Unlike our method, these baselines require access to the full test set, whereas ours supports adaptive, causal anomaly detection without full-dataset access.

### C.2.2 METRICS

**Threshold-dependent metrics.** We follow the evaluation pipeline provided in Liu & Paparrizos (2024). Given anomaly scores $\{\beta_i \in [0, 1]\}_{i=1}^t$ (interpreted as $p$-values, where smaller values indicate stronger outliers) and ground-truth labels $\{\ell_i \in \{0, 1\}\}_{i=1}^t$, we evaluate metrics $M(\{\ell_i\}, \{\hat{\ell}_i\}) \in [0, 1]$ where larger is better. Examples include Affiliation-F and PA-F1. For a family of thresholds $\{\alpha_j \in [0, 1]\}_{j=1}^k$, we select the best score

$$j^* = \arg\max_{j \in [k]} M\Big(\{\ell_i\}, \{\mathbf{1}[\beta_i \leq \alpha_j]\}\Big), \quad M^* = M\Big(\{\ell_i\}, \{\mathbf{1}[\beta_i \leq \alpha_{j^*}]\}\Big),$$

and report the corresponding false positive rate $FPR(\alpha_{j^*}) = \mathbb{P}[\beta_i \leq \alpha_{j^*} \mid \ell_i = 0]$, as well as the calibration error $CalErr(\alpha_{j^*}) = |FPR(\alpha_{j^*}) - \alpha_{j^*}|$. Thresholds are evaluated on a uniform grid (linspace) with finer resolution at small $p$-values: 21 values in $[0.001, 0.01]$, 21 values in $[0.02, 0.1]$, and 21 values in $[0.2, 1]$.

**Threshold-independent metrics.** We also report AUC and VUS-PR. In both cases, integration is performed using 250 quantiles of each method's calibration score distribution.

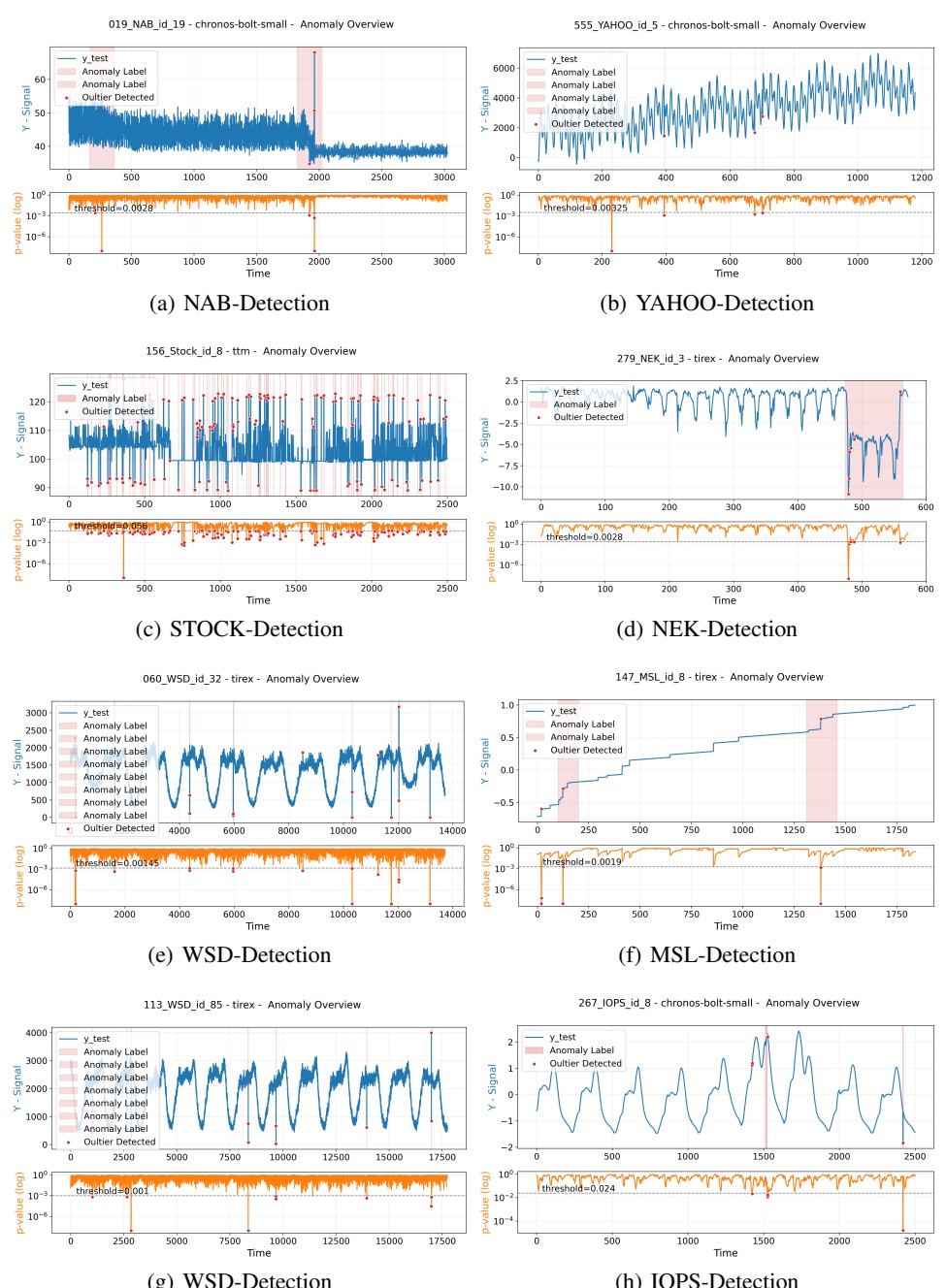

Figure 6: Example signals (blue) with ground-truth anomaly labels (red areas), detected outliers (red dots) occur when adaptive $p$-values (orange) fall below a threshold under our proposed $\mathcal{W}_1$-ACAS method.

### C.2.3 ADDITIONAL RESULTS

Figure 6 provides additional detection examples, while Figure 7 illustrates the trade-offs between FPR and F1 scores (PA-F1 and Affiliation-F) at the operating points that maximize the respective F1 metric, as defined in Appendix C.2.2. For PA-F1, $\mathcal{W}_1$-ACAS consistently dominates competing approaches. For Affiliation-F, $\mathcal{W}_1$-ACAS yields operating points that are rarely dominated and is the top-performing method in several datasets.

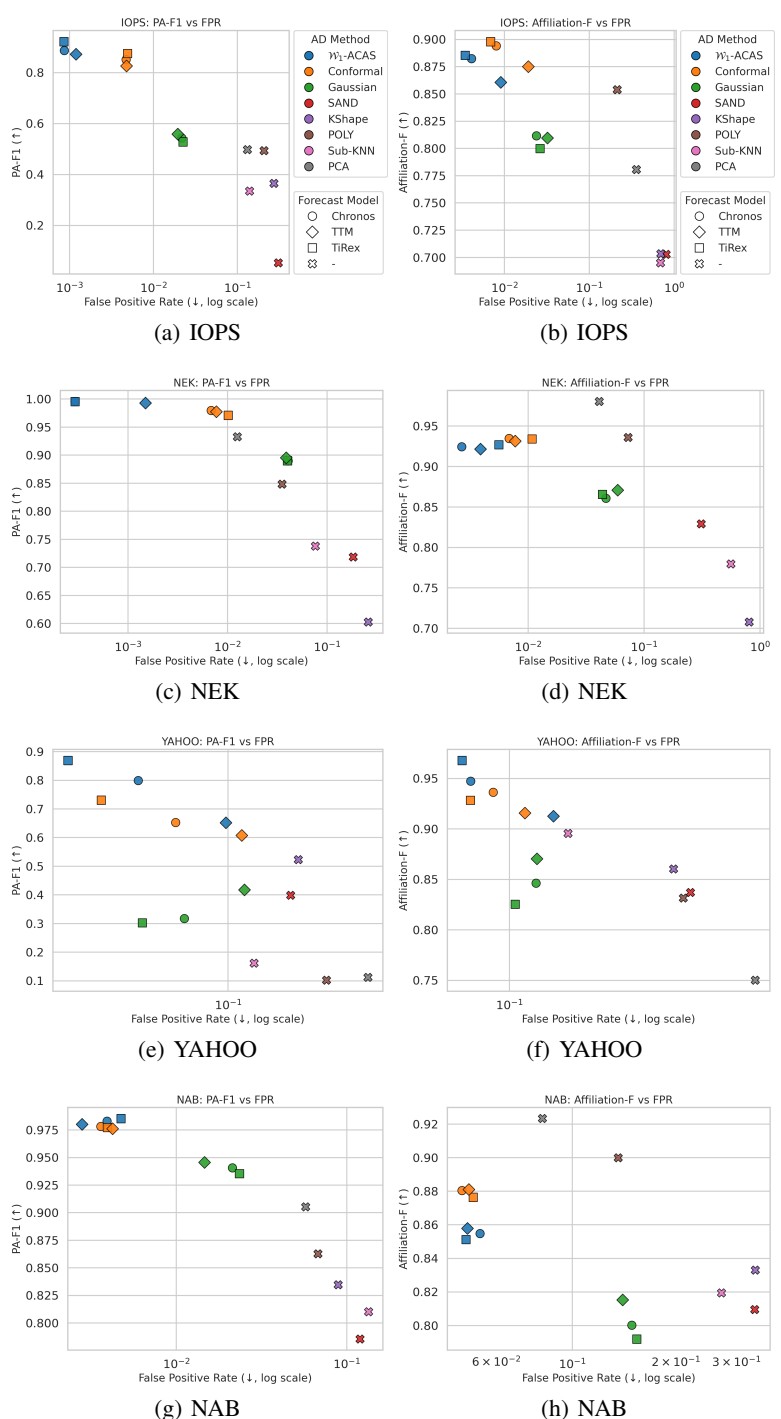

Figure 7: **Trade-offs between false positive rate and detection performance across datasets.** Left column: PA-F1 vs FPR (log scale). Right column: Affiliation-F vs FPR (log scale). Each point uses color for AD method and marker for forecast model. The operating points of $\mathcal{W}_1$-ACAS (blue), in most cases, achieve both the highest F1 score and lowest FPR, especially for PA-F1. Within the same TSFM model, $\mathcal{W}_1$-ACAS is better than the alternatives, and in general dominate most of the alternatives.

**Hyperparameter Sensitivity.** Figure 8 examines the effect of aggregating different numbers of forecast horizons. Performance generally stabilizes once more than 10 horizons are included, with

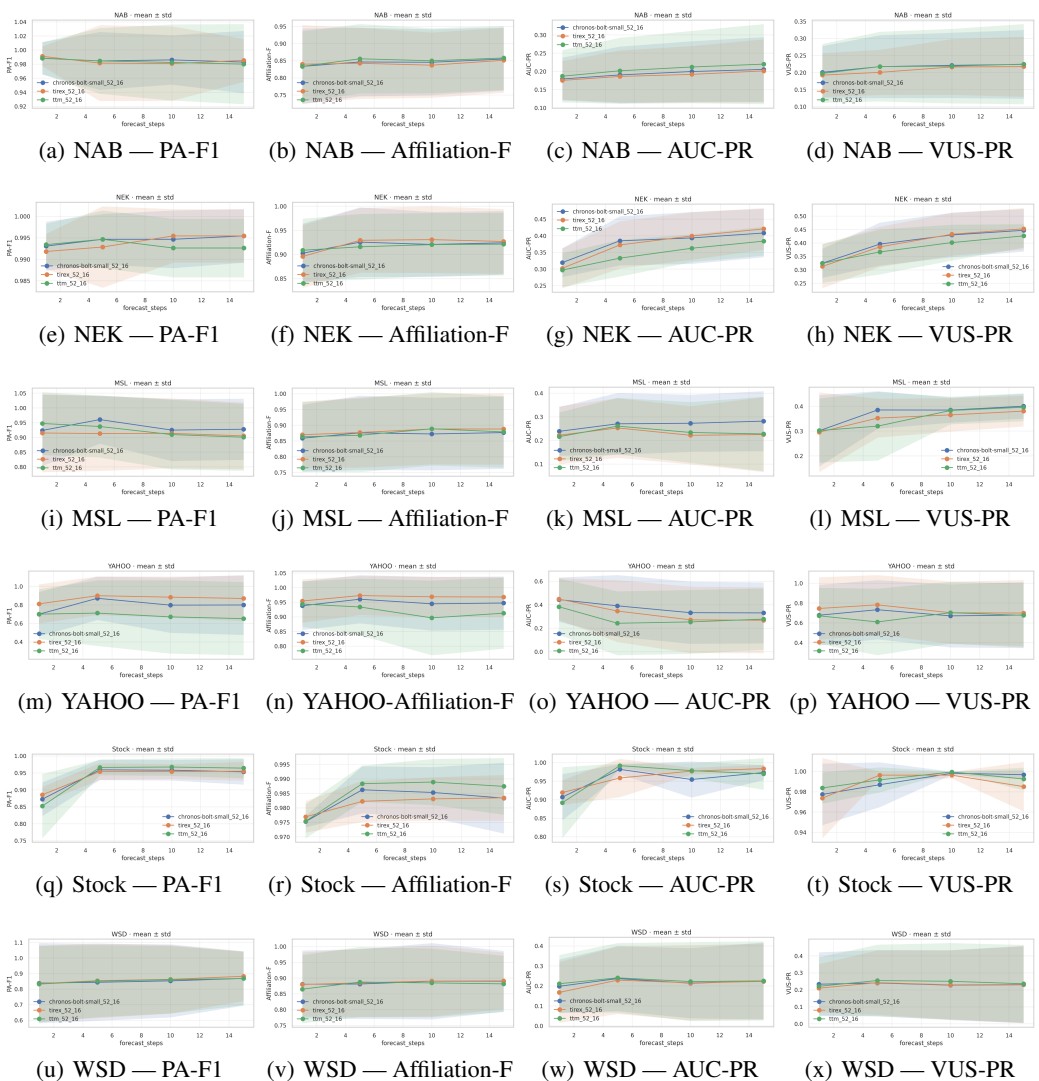

Figure 8: Performance of $\mathcal{W}_1$-ACAS when aggregating different forecast steps. Rows correspond to datasets (NAB, NEK, MSL, YAHOO, Stock, WSD) and columns to metrics (PA-F1, Affiliation-F, AUC-PR, VUS-PR).

limited gains beyond this point. Figure 9 shows the sensitivity of $\mathcal{W}_1$-ACAS to the learning rate $\gamma$ (with $\alpha_c = 0.01$ and $n_b = 10$). Since the weights are updated using ADAM, $\gamma$ must remain sufficiently small; empirically, the method exhibits no significant variability for small learning rates. Figure 10 illustrates the effect of the batch size $n_b$ (with $\gamma = 0.001$ and $\alpha_c = 0.01$). This parameter controls the number of samples used in the Wasserstein distance computation: if the distribution of nonconformity scores changes over time, $n_b$ should not be too large. In practice, the method is only mildly sensitive to $n_b$, with smaller values performing slightly better on some datasets. Finally, Figure 11 examines the sensitivity to the critical alarm rate $\alpha_c$. This parameter determines the maximum acceptable $p$-value resolution: smaller values require a larger number of in-distribution past observations $n_c$ for stable quantile estimation, but do not impose a lower bound on the smallest detectable anomaly level.

**Per-dataset performance** Tables 2, 3, and 4 report per-dataset metrics, which align with and reinforce the trends discussed in the main Experimental section. Table 5 summarizes the forecasting performance of the TSFM models across datasets. Overall, the models exhibit broadly similar MAE/RMSE values, which aligns with their comparable anomaly-detection performance once fore-

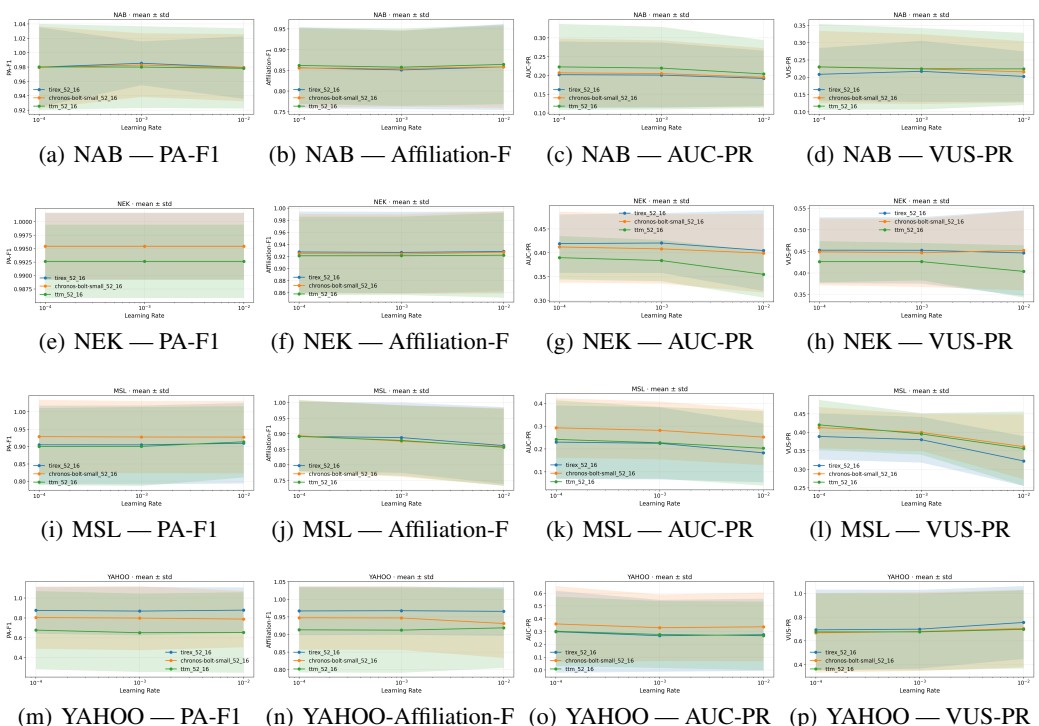

Figure 9: Performance of $\mathcal{W}_1$-ACAS when aggregating different learning rate. Rows correspond to datasets (NAB, NEK, MSL, YAHOO, Stock, WSD) and columns to metrics (PA-F1, Affiliation-F, AUC-PR, VUS-PR).

cast errors are properly calibrated online using $\mathcal{W}_1$-ACAS. Notably, the slightly higher forecasting error of TTM on YAHOO corresponds to its lower anomaly-detection performance in Table 2, suggesting a consistent relationship between forecast quality and downstream AD results.

**Computation Time.** The average per-sample computation time of $\mathcal{W}_1$-ACAS with a 15-step forecast is $0.025 \pm 0.012$ seconds per sample per feature on a single V100 32 GB GPU. Note that this implementation updates weights for all 15 predictors serially, these updates are independent and can be parallelized to further reduce runtime.

### C.2.4 EXTENSION TO MULTIVARIATE TIME SERIES ANOMALY DETECTION.

**$\mathcal{W}_1$-ACAS via $p$-value aggregation.** . Lets consider a multivariate time series with features $f \in [n_f]$, we can run Algorithm 1 independently on each dimension to obtain per-feature $p$-values $\bar{\beta}^f t + 1$ at time $t + 1$ (as defined in Eq. 15). These are then combined into a single anomaly score using standard $p$-value combination methods Heard & Rubin-Delanchy (2018):

- Fisher's Method (Fisher, 1970): Combined p-value is $\rho_{t+1} = 1 - F_{\chi^2_{2n_f}}^{-1}(Z_{t+1})$ with
  $Z_{t+1} = -2 \sum_f \bar{\beta}^f_{t+1}$.

- Harmonic Mean $p$-value (HMP) (Wilson, 2019): Combined p-value is $\rho_{t+1} = \frac{n_f}{\sum_f 1/\bar{\beta}^f_{t+1}}$.

We refer to these variants as $\mathcal{W}_1$-ACAS-F and $\mathcal{W}_1$-ACAS-H, respectively.

**Experiments and Results.** We adopt the curated subsets from the TSB-AD benchmark (Liu & Paparrizos, 2024): TAO (Laboratory, 2024) (13 curated series, each with ~10k samples and 3 features, containing both sequential and point anomalies), GECCO (Rehbach et al., 2018) (a single long sequence with 9 features and over 138k samples), Genesis (von Birgelen & Niggemann, 2018)

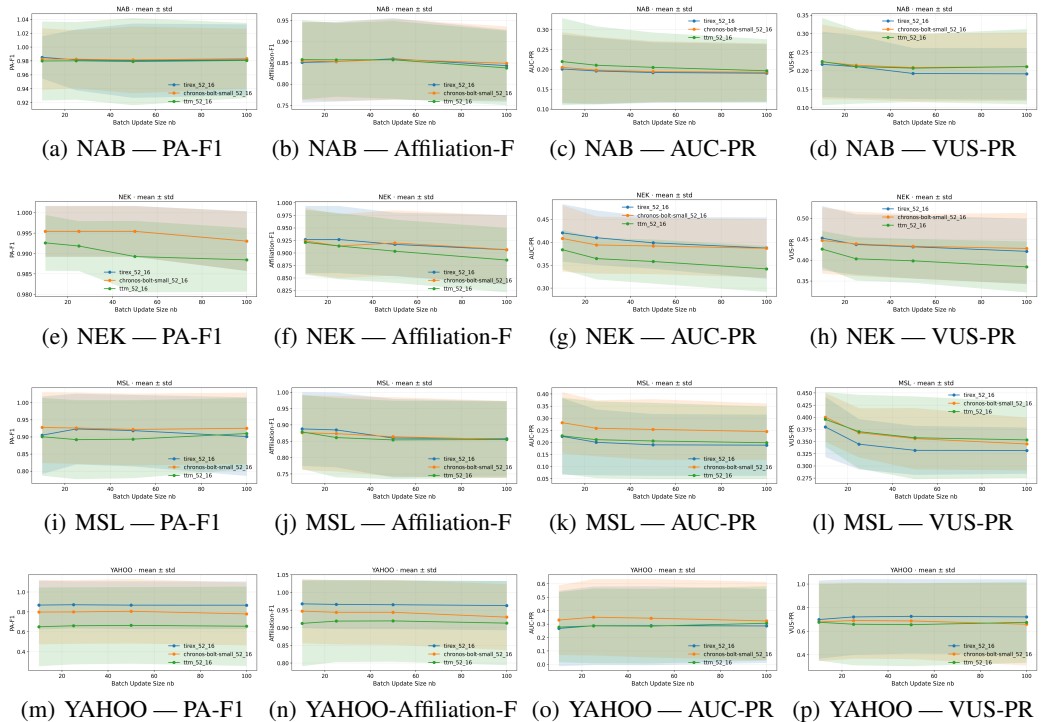

Figure 10: Performance of $\mathcal{W}_1$-ACAS when aggregating different batch size update $n_b$. Rows correspond to datasets (NAB, NEK, MSL, YAHOO, Stock, WSD) and columns to metrics (PA-F1, Affiliation-F, AUC-PR, VUS-PR).

(1 sequence with 18 features and over 16k samples), and LTDB (Goldberger et al., 2000) (5 curated sequences, each with 2 features and approximately 100k samples).

We evaluate our multivariate extensions, $\mathcal{W}_1$-ACAS-F and $\mathcal{W}_1$-ACAS-H, combined with Chronos and TiRex forecasters that leverage all available historical context (up to their maximum context window, with a minimum of 52 past points). These are compared against strong semi-supervised deep anomaly detection baselines (Liu & Paparrizos, 2024): CNN (Munir et al., 2018), Omni-Anomaly (Su et al., 2019), and USAD (Audibert et al., 2020), which benefit from being trained directly on non-anomalous segments. As reported in Table 6, both $\mathcal{W}_1$-ACAS-F and $\mathcal{W}_1$-ACAS-H achieve the best or highly competitive performance across all multivariate datasets, demonstrating the effectiveness of our $p$-value aggregation extension in this setting.

## D    THE USE OF LARGE LANGUAGE MODELS (LLMS)

We used large language models (LLMs) to assist with improving the readability and clarity of the manuscript. LLMs were used to improve and summarize the language in certain paragraphs, and to refine code for generating plots.

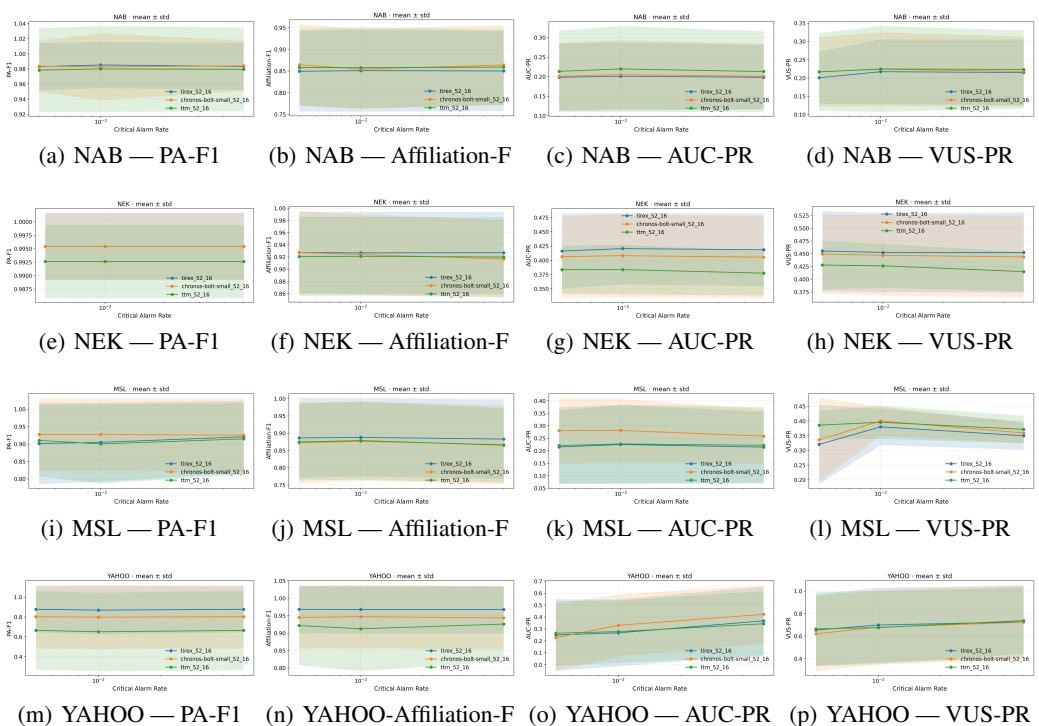

(a) NAB — PA-F1    (b) NAB — Affiliation-F    (c) NAB — AUC-PR    (d) NAB — VUS-PR

(e) NEK — PA-F1    (f) NEK — Affiliation-F    (g) NEK — AUC-PR    (h) NEK — VUS-PR

(i) MSL — PA-F1    (j) MSL — Affiliation-F    (k) MSL — AUC-PR    (l) MSL — VUS-PR

(m) YAHOO — PA-F1    (n) YAHOO-Affiliation-F    (o) YAHOO — AUC-PR    (p) YAHOO — VUS-PR

Figure 11: Performance of $\mathcal{W}_1$-ACAS when aggregating different critical alarm rate $\alpha_c$. Rows correspond to datasets (NAB, NEK, MSL, YAHOO, Stock, WSD) and columns to metrics (PA-F1, Affiliation-F, AUC-PR, VUS-PR).

| Dataset | Forecaster | AD Model | PA-F1 ↑ | Affiliation-F ↑ | FPR ↓ | CalErr ↓ | AUC-PR ↑ | VUC-PR ↑ |
|---|---|---|---|---|---|---|---|---|
| YAHOO | - | KShapeAD | 0.523 ± 0.430 | 0.860 ± 0.151 | 0.359 ± 0.437 | 0.119 ± 0.183 | 0.036 ± 0.110 | 0.220 ± 0.225 |
| YAHOO | - | POLY | 0.102 ± 0.217 | 0.831 ± 0.126 | 0.387 ± 0.367 | 0.244 ± 0.240 | 0.037 ± 0.127 | 0.139 ± 0.125 |
| YAHOO | - | Sub-KNN | 0.161 ± 0.273 | 0.895 ± 0.109 | 0.158 ± 0.222 | 0.157 ± 0.161 | 0.016 ± 0.043 | 0.260 ± 0.197 |
| YAHOO | - | Sub-PCA | 0.112 ± 0.261 | 0.750 ± 0.115 | 0.677 ± 0.410 | 0.099 ± 0.163 | 0.056 ± 0.134 | 0.125 ± 0.199 |
| YAHOO | - | SAND | 0.398 ± 0.416 | 0.837 ± 0.147 | 0.409 ± 0.434 | 0.097 ± 0.114 | 0.024 ± 0.071 | 0.198 ± 0.180 |
| YAHOO | - | CNN* | 0.596 ± 0.438 | 0.853 ± 0.146 | 0.242 ± 0.407 | 0.240 ± 0.321 | 0.053 ± 0.147 | 0.160 ± 0.258 |
| YAHOO | - | OmniAnomaly* | 0.272 ± 0.381 | 0.791 ± 0.136 | 0.384 ± 0.446 | 0.313 ± 0.318 | 0.195 ± 0.255 | 0.351 ± 0.378 |
| YAHOO | - | USAD* | 0.113 ± 0.287 | 0.736 ± 0.098 | 0.610 ± 0.381 | 0.154 ± 0.187 | 0.068 ± 0.160 | 0.201 ± 0.288 |
| YAHOO | - | MOMENT_ZS | 0.134 ± 0.222 | 0.832 ± 0.121 | 0.215 ± 0.325 | 0.195 ± 0.193 | 0.086 ± 0.188 | 0.233 ± 0.235 |
| YAHOO | Chronos | $\mathcal{W}_1$-ACAS | 0.798 ± 0.323 | 0.947 ± 0.091 | 0.074 ± 0.253 | 0.007 ± 0.017 | **0.330 ± 0.259** | 0.679 ± 0.332 |
| YAHOO | Chronos | conformal | 0.652 ± 0.361 | 0.936 ± 0.091 | 0.088 ± 0.258 | 0.015 ± 0.028 | 0.147 ± 0.224 | 0.485 ± 0.347 |
| YAHOO | Chronos | gaussian | 0.317 ± 0.284 | 0.846 ± 0.098 | 0.123 ± 0.265 | 0.028 ± 0.100 | 0.028 ± 0.100 | 0.511 ± 0.345 |
| YAHOO | Tirex | $\mathcal{W}_1$-ACAS | **0.869 ± 0.244** | **0.968 ± 0.068** | 0.069 ± 0.253 | **0.003 ± 0.007** | 0.267 ± 0.280 | **0.699 ± 0.331** |
| YAHOO | Tirex | conformal | 0.730 ± 0.310 | 0.928 ± 0.091 | 0.074 ± 0.252 | 0.009 ± 0.015 | 0.176 ± 0.259 | 0.559 ± 0.317 |
| YAHOO | Tirex | gaussian | 0.302 ± 0.269 | 0.825 ± 0.101 | 0.105 ± 0.252 | 0.041 ± 0.062 | 0.030 ± 0.114 | 0.546 ± 0.310 |
| YAHOO | TTM | $\mathcal{W}_1$-ACAS | 0.651 ± 0.395 | 0.912 ± 0.121 | 0.141 ± 0.343 | 0.034 ± 0.057 | 0.277 ± 0.261 | 0.676 ± 0.324 |
| YAHOO | TTM | conformal | 0.607 ± 0.413 | 0.916 ± 0.108 | 0.113 ± 0.273 | 0.052 ± 0.082 | 0.172 ± 0.230 | 0.611 ± 0.350 |
| YAHOO | TTM | gaussian | 0.417 ± 0.334 | 0.870 ± 0.113 | 0.124 ± 0.276 | 0.050 ± 0.104 | 0.028 ± 0.099 | 0.560 ± 0.331 |
| NEK | - | KShapeAD | 0.602 ± 0.292 | 0.708 ± 0.043 | 0.807 ± 0.284 | 0.077 ± 0.118 | 0.216 ± 0.179 | 0.152 ± 0.138 |
| NEK | - | POLY | 0.848 ± 0.149 | 0.936 ± 0.066 | 0.073 ± 0.058 | 0.478 ± 0.135 | 0.063 ± 0.073 | 0.616 ± 0.162 |
| NEK | - | Sub-KNN | 0.738 ± 0.307 | 0.779 ± 0.098 | 0.561 ± 0.451 | 0.054 ± 0.086 | 0.172 ± 0.068 | 0.321 ± 0.131 |
| NEK | - | Sub-PCA | 0.933 ± 0.107 | **0.980 ± 0.022** | 0.041 ± 0.068 | 0.393 ± 0.190 | 0.007 ± 0.013 | 0.705 ± 0.230 |
| NEK | - | SAND | 0.718 ± 0.340 | 0.829 ± 0.105 | 0.312 ± 0.338 | 0.201 ± 0.135 | 0.325 ± 0.196 | 0.214 ± 0.203 |
| NEK | - | CNN* | 0.996 ± 0.006 | 0.965 ± 0.078 | **0.000 ± 0.000** | 0.859 ± 0.046 | **0.438 ± 0.197** | 0.730 ± 0.218 |
| NEK | - | OmniAnomaly* | **0.998 ± 0.005** | 0.968 ± 0.077 | 0.001 ± 0.001 | 0.875 ± 0.033 | 0.195 ± 0.189 | **0.872 ± 0.132** |
| NEK | - | USAD* | 0.785 ± 0.295 | 0.933 ± 0.058 | 0.179 ± 0.162 | 0.440 ± 0.124 | 0.006 ± 0.015 | 0.555 ± 0.174 |
| NEK | - | MOMENT_ZS | 0.849 ± 0.200 | 0.942 ± 0.028 | 0.125 ± 0.095 | 0.496 ± 0.165 | 0.046 ± 0.041 | 0.583 ± 0.138 |
| NEK | Chronos | $\mathcal{W}_1$-ACAS | 0.995 ± 0.006 | 0.924 ± 0.066 | 0.003 ± 0.005 | **0.004 ± 0.003** | 0.408 ± 0.073 | 0.447 ± 0.079 |
| NEK | Chronos | conformal | 0.979 ± 0.012 | 0.934 ± 0.067 | 0.007 ± 0.006 | 0.007 ± 0.004 | 0.418 ± 0.104 | 0.490 ± 0.092 |
| NEK | Chronos | gaussian | 0.890 ± 0.021 | 0.860 ± 0.069 | 0.047 ± 0.028 | 0.045 ± 0.025 | 0.347 ± 0.054 | 0.519 ± 0.093 |
| NEK | Tirex | $\mathcal{W}_1$-ACAS | 0.995 ± 0.006 | 0.927 ± 0.067 | 0.006 ± 0.015 | 0.005 ± 0.003 | 0.421 ± 0.063 | 0.453 ± 0.077 |
| NEK | Tirex | conformal | 0.971 ± 0.011 | 0.934 ± 0.066 | 0.011 ± 0.007 | 0.009 ± 0.004 | 0.421 ± 0.097 | 0.496 ± 0.097 |
| NEK | Tirex | gaussian | 0.890 ± 0.027 | 0.865 ± 0.064 | 0.044 ± 0.021 | 0.043 ± 0.021 | 0.354 ± 0.056 | 0.513 ± 0.099 |
| NEK | TTM | $\mathcal{W}_1$-ACAS | 0.993 ± 0.007 | 0.921 ± 0.065 | 0.004 ± 0.007 | 0.005 ± 0.003 | 0.384 ± 0.043 | 0.426 ± 0.043 |
| NEK | TTM | conformal | 0.977 ± 0.016 | 0.931 ± 0.067 | 0.008 ± 0.007 | 0.008 ± 0.005 | 0.417 ± 0.047 | 0.471 ± 0.057 |
| NEK | TTM | gaussian | 0.895 ± 0.012 | 0.871 ± 0.068 | 0.059 ± 0.045 | 0.040 ± 0.019 | 0.337 ± 0.060 | 0.501 ± 0.064 |

Table 2: **Performance Summary per datasets.** Entries indicate the mean ± standard deviation computed by averaging within each dataset group. Higher numbers are better for PA-F1, Affiliation-F, AUC-PR, VUS-PR; lower numbers are better for FPR, and calibration error (CalErr). Methods marked with * denote deep learning semi-supervised approaches; the best overall method is shown in **bold**, and the best non–semi-supervised method is underlined when different from the bold one.

| Dataset | Forecaster | AD Model | PA-F1 ↑ | Affiliation-F ↑ | FPR ↓ | CalErr ↓ | AUC-PR ↑ | VUC-PR ↑ |
|---|---|---|---|---|---|---|---|---|
| MSL | - | KShapeAD | 0.854 ± 0.207 | 0.915 ± 0.113 | 0.116 ± 0.154 | 0.188 ± 0.131 | 0.108 ± 0.119 | 0.260 ± 0.163 |
| MSL | - | POLY | 0.619 ± 0.330 | 0.881 ± 0.114 | 0.248 ± 0.339 | 0.076 ± 0.108 | 0.077 ± 0.106 | 0.353 ± 0.187 |
| MSL | - | Sub-KNN | 0.685 ± 0.379 | 0.835 ± 0.124 | 0.293 ± 0.411 | 0.137 ± 0.099 | 0.132 ± 0.164 | 0.179 ± 0.153 |
| MSL | - | Sub-PCA | 0.683 ± 0.354 | 0.882 ± 0.110 | 0.175 ± 0.248 | 0.145 ± 0.185 | 0.056 ± 0.071 | 0.371 ± 0.329 |
| MSL | - | SAND | 0.655 ± 0.328 | 0.877 ± 0.122 | 0.242 ± 0.251 | 0.176 ± 0.153 | 0.064 ± 0.068 | 0.303 ± 0.179 |
| MSL | - | CNN* | 0.826 ± 0.225 | 0.885 ± 0.099 | 0.096 ± 0.217 | 0.460 ± 0.407 | 0.105 ± 0.105 | 0.308 ± 0.264 |
| MSL | - | OmniAnomaly* | 0.818 ± 0.257 | 0.879 ± 0.106 | 0.038 ± 0.063 | 0.588 ± 0.409 | 0.121 ± 0.133 | 0.344 ± 0.262 |
| MSL | - | USAD* | 0.667 ± 0.349 | 0.881 ± 0.108 | 0.133 ± 0.197 | 0.384 ± 0.288 | 0.060 ± 0.095 | 0.415 ± 0.389 |
| MSL | - | MOMENT_ZS | 0.799 ± 0.300 | **0.905 ± 0.128** | 0.151 ± 0.375 | 0.429 ± 0.328 | 0.134 ± 0.093 | **0.501 ± 0.290** |
| MSL | Chronos | $\mathcal{W}_1$-ACAS | **0.928 ± 0.104** | 0.876 ± 0.115 | 0.033 ± 0.062 | 0.022 ± 0.027 | 0.282 ± 0.127 | 0.400 ± 0.050 |
| MSL | Chronos | conformal | 0.829 ± 0.318 | 0.813 ± 0.122 | 0.310 ± 0.472 | 0.159 ± 0.370 | 0.308 ± 0.159 | 0.306 ± 0.175 |
| MSL | Chronos | gaussian | 0.854 ± 0.126 | 0.842 ± 0.101 | 0.180 ± 0.363 | 0.074 ± 0.151 | 0.262 ± 0.124 | 0.368 ± 0.157 |
| MSL | Tirex | $\mathcal{W}_1$-ACAS | 0.905 ± 0.113 | 0.888 ± 0.113 | 0.152 ± 0.374 | **0.017 ± 0.020** | 0.225 ± 0.158 | 0.380 ± 0.062 |
| MSL | Tirex | conformal | 0.826 ± 0.315 | 0.816 ± 0.121 | 0.312 ± 0.471 | 0.158 ± 0.371 | 0.226 ± 0.160 | 0.299 ± 0.171 |
| MSL | Tirex | gaussian | 0.856 ± 0.124 | 0.841 ± 0.100 | 0.187 ± 0.362 | 0.131 ± 0.182 | 0.267 ± 0.118 | 0.378 ± 0.170 |
| MSL | TTM | $\mathcal{W}_1$-ACAS | 0.901 ± 0.114 | 0.879 ± 0.113 | **0.023 ± 0.039** | 0.017 ± 0.026 | 0.227 ± 0.157 | 0.396 ± 0.056 |
| MSL | TTM | conformal | 0.803 ± 0.308 | 0.812 ± 0.120 | 0.319 ± 0.467 | 0.165 ± 0.368 | 0.243 ± 0.196 | 0.396 ± 0.181 |
| MSL | TTM | gaussian | 0.855 ± 0.118 | 0.849 ± 0.104 | 0.180 ± 0.364 | 0.074 ± 0.133 | 0.286 ± 0.174 | 0.416 ± 0.122 |
| NAB | - | KShapeAD | 0.835 ± 0.222 | 0.833 ± 0.141 | 0.341 ± 0.424 | 0.195 ± 0.250 | 0.123 ± 0.162 | 0.272 ± 0.220 |
| NAB | - | POLY | 0.863 ± 0.203 | 0.900 ± 0.109 | 0.136 ± 0.251 | 0.232 ± 0.248 | 0.087 ± 0.073 | 0.322 ± 0.189 |
| NAB | - | Sub-KNN | 0.810 ± 0.257 | 0.819 ± 0.124 | 0.272 ± 0.383 | 0.273 ± 0.318 | 0.153 ± 0.150 | 0.304 ± 0.283 |
| NAB | - | Sub-PCA | 0.905 ± 0.185 | 0.923 ± 0.100 | 0.082 ± 0.223 | 0.341 ± 0.320 | 0.199 ± 0.245 | 0.427 ± 0.286 |
| NAB | - | SAND | 0.785 ± 0.245 | 0.809 ± 0.130 | 0.340 ± 0.422 | 0.151 ± 0.140 | 0.130 ± 0.131 | 0.296 ± 0.207 |
| NAB | - | CNN* | 0.982 ± 0.054 | **0.937 ± 0.077** | **0.008 ± 0.034** | 0.468 ± 0.420 | 0.194 ± 0.115 | 0.260 ± 0.146 |
| NAB | - | OmniAnomaly* | 0.977 ± 0.082 | 0.925 ± 0.091 | 0.068 ± 0.231 | 0.506 ± 0.366 | 0.201 ± 0.071 | 0.274 ± 0.139 |
| NAB | - | USAD* | 0.927 ± 0.139 | 0.926 ± 0.101 | 0.123 ± 0.263 | 0.473 ± 0.314 | 0.200 ± 0.206 | **0.445 ± 0.237** |
| NAB | - | MOMENT_ZS | 0.958 ± 0.115 | 0.931 ± 0.103 | 0.129 ± 0.315 | 0.490 ± 0.341 | 0.220 ± 0.218 | 0.407 ± 0.216 |
| NAB | Chronos | $\mathcal{W}_1$-ACAS | 0.983 ± 0.044 | 0.855 ± 0.092 | 0.054 ± 0.208 | 0.012 ± 0.028 | 0.205 ± 0.089 | 0.224 ± 0.101 |
| NAB | Chronos | conformal | 0.978 ± 0.057 | 0.880 ± 0.089 | 0.048 ± 0.208 | 0.013 ± 0.011 | 0.205 ± 0.094 | 0.232 ± 0.108 |
| NAB | Chronos | gaussian | 0.941 ± 0.051 | 0.800 ± 0.103 | 0.149 ± 0.328 | 0.077 ± 0.097 | 0.201 ± 0.094 | 0.223 ± 0.098 |
| NAB | Tirex | $\mathcal{W}_1$-ACAS | **0.985 ± 0.030** | 0.851 ± 0.095 | 0.049 ± 0.208 | 0.011 ± 0.027 | 0.201 ± 0.087 | 0.217 ± 0.088 |
| NAB | Tirex | conformal | 0.977 ± 0.057 | 0.876 ± 0.090 | 0.052 ± 0.207 | 0.012 ± 0.017 | 0.201 ± 0.097 | 0.228 ± 0.096 |
| NAB | Tirex | gaussian | 0.935 ± 0.059 | 0.792 ± 0.095 | 0.154 ± 0.331 | 0.085 ± 0.105 | 0.195 ± 0.097 | 0.218 ± 0.086 |
| NAB | TTM | $\mathcal{W}_1$-ACAS | 0.980 ± 0.057 | 0.858 ± 0.093 | 0.050 ± 0.207 | **0.010 ± 0.019** | **0.220 ± 0.110** | 0.225 ± 0.118 |
| NAB | TTM | conformal | 0.976 ± 0.058 | 0.881 ± 0.087 | 0.050 ± 0.207 | 0.008 ± 0.011 | 0.217 ± 0.115 | 0.231 ± 0.125 |
| NAB | TTM | gaussian | 0.945 ± 0.064 | 0.815 ± 0.103 | 0.140 ± 0.310 | 0.078 ± 0.109 | 0.203 ± 0.104 | 0.235 ± 0.124 |

Table 3: **Performance Summary per datasets.** Entries indicate the mean ± standard deviation computed by averaging within each dataset group. Higher numbers are better for PA-F1, Affiliation-F, AUC-PR, VUS-PR; lower numbers are better for FPR, and calibration error (CalErr). Methods marked with * denote deep learning semi-supervised approaches; the best overall method is shown in **bold**, and the best non–semi-supervised method is underlined when different from the bold one.

| Dataset | Forecaster | AD Model | PA-F1 ↑ | Affiliation-F ↑ | FPR ↓ | CalErr ↓ | AUC-PR ↑ | VUC-PR ↑ |
|---|---|---|---|---|---|---|---|---|
| WSD | - | KShapeAD | 0.117 ± 0.210 | 0.722 ± 0.084 | 0.469 ± 0.361 | 0.162 ± 0.133 | 0.011 ± 0.023 | 0.061 ± 0.116 |
| WSD | - | POLY | 0.475 ± 0.337 | 0.862 ± 0.138 | 0.199 ± 0.333 | 0.281 ± 0.240 | 0.006 ± 0.010 | 0.226 ± 0.223 |
| WSD | - | Sub-KNN | 0.195 ± 0.237 | 0.755 ± 0.088 | 0.312 ± 0.422 | 0.054 ± 0.071 | 0.026 ± 0.066 | 0.103 ± 0.135 |
| WSD | - | Sub-PCA | 0.208 ± 0.296 | 0.747 ± 0.093 | 0.479 ± 0.393 | 0.205 ± 0.212 | 0.040 ± 0.110 | 0.102 ± 0.135 |
| WSD | - | CNN* | **0.980 ± 0.038** | **0.970 ± 0.061** | **0.001 ± 0.001** | 0.712 ± 0.287 | 0.033 ± 0.035 | 0.216 ± 0.200 |
| WSD | - | OmniAnomaly* | 0.414 ± 0.431 | 0.804 ± 0.134 | 0.471 ± 0.470 | 0.328 ± 0.353 | 0.047 ± 0.116 | 0.090 ± 0.116 |
| WSD | - | USAD* | 0.102 ± 0.210 | 0.711 ± 0.061 | 0.602 ± 0.335 | 0.269 ± 0.209 | 0.009 ± 0.011 | 0.041 ± 0.059 |
| WSD | - | MOMENT_ZS | 0.568 ± 0.238 | 0.944 ± 0.078 | 0.059 ± 0.194 | 0.504 ± 0.284 | 0.030 ± 0.061 | **0.394 ± 0.248** |
| WSD | Chronos | $\mathcal{W}_1$-ACAS | 0.868 ± 0.175 | 0.890 ± 0.096 | 0.096 ± 0.292 | 0.007 ± 0.016 | 0.224 ± 0.192 | 0.230 ± 0.226 |
| WSD | Chronos | conformal | 0.810 ± 0.193 | 0.882 ± 0.086 | 0.098 ± 0.292 | 0.006 ± 0.009 | 0.105 ± 0.120 | 0.227 ± 0.172 |
| WSD | Chronos | gaussian | 0.387 ± 0.207 | 0.788 ± 0.072 | 0.111 ± 0.283 | 0.025 ± 0.025 | 0.079 ± 0.085 | 0.226 ± 0.173 |
| WSD | Tirex | $\mathcal{W}_1$-ACAS | 0.882 ± 0.159 | 0.891 ± 0.090 | 0.048 ± 0.210 | 0.007 ± 0.022 | 0.222 ± 0.190 | 0.239 ± 0.228 |
| WSD | Tirex | conformal | 0.841 ± 0.173 | 0.886 ± 0.087 | 0.052 ± 0.222 | 0.006 ± 0.006 | 0.119 ± 0.115 | 0.238 ± 0.208 |
| WSD | Tirex | gaussian | 0.393 ± 0.210 | 0.783 ± 0.074 | 0.110 ± 0.283 | 0.023 ± 0.023 | 0.067 ± 0.074 | 0.231 ± 0.202 |
| WSD | TTM | $\mathcal{W}_1$-ACAS | 0.868 ± 0.172 | 0.882 ± 0.089 | 0.064 ± 0.228 | **0.005 ± 0.012** | **0.225 ± 0.198** | 0.236 ± 0.229 |
| WSD | TTM | conformal | 0.812 ± 0.174 | 0.879 ± 0.084 | 0.067 ± 0.229 | 0.007 ± 0.006 | 0.191 ± 0.146 | 0.237 ± 0.194 |
| WSD | TTM | gaussian | 0.389 ± 0.212 | 0.782 ± 0.076 | 0.112 ± 0.292 | 0.026 ± 0.032 | 0.063 ± 0.066 | 0.230 ± 0.196 |
| Stock | - | KShapeAD | 0.135 ± 0.072 | 0.680 ± 0.010 | 0.951 ± 0.095 | 0.081 ± 0.060 | 0.060 ± 0.036 | 0.603 ± 0.342 |
| Stock | - | POLY | 0.201 ± 0.089 | 0.720 ± 0.082 | 0.805 ± 0.360 | 0.245 ± 0.248 | 0.000 ± 0.000 | 0.615 ± 0.350 |
| Stock | - | Sub-KNN | 0.150 ± 0.087 | 0.678 ± 0.008 | 0.979 ± 0.027 | 0.175 ± 0.136 | 0.083 ± 0.067 | 0.627 ± 0.369 |
| Stock | - | Sub-PCA | 0.199 ± 0.086 | 0.726 ± 0.090 | 0.792 ± 0.335 | 0.128 ± 0.199 | 0.117 ± 0.068 | 0.844 ± 0.087 |
| Stock | - | SAND | 0.174 ± 0.100 | 0.687 ± 0.001 | 0.933 ± 0.086 | 0.137 ± 0.192 | 0.071 ± 0.027 | 0.549 ± 0.549 |
| Stock | - | CNN* | **0.996 ± 0.003** | **0.999 ± 0.001** | **0.001 ± 0.000** | 0.872 ± 0.066 | 0.900 ± 0.105 | 0.980 ± 0.027 |
| Stock | - | OmniAnomaly* | 0.372 ± 0.038 | 0.886 ± 0.046 | 0.242 ± 0.122 | 0.688 ± 0.160 | 0.284 ± 0.054 | 0.962 ± 0.026 |
| Stock | - | USAD* | 0.146 ± 0.070 | 0.676 ± 0.008 | 0.983 ± 0.019 | 0.021 ± 0.027 | 0.068 ± 0.043 | 0.747 ± 0.149 |
| Stock | - | MOMENT_ZS | 0.163 ± 0.071 | 0.680 ± 0.006 | 0.931 ± 0.045 | 0.049 ± 0.024 | 0.093 ± 0.051 | 0.598 ± 0.365 |
| Stock | Chronos | $\mathcal{W}_1$-ACAS | 0.959 ± 0.031 | 0.985 ± 0.008 | 0.009 ± 0.012 | 0.074 ± 0.040 | 0.973 ± 0.020 | **0.998 ± 0.001** |
| Stock | Chronos | conformal | 0.959 ± 0.039 | 0.990 ± 0.009 | 0.009 ± 0.010 | 0.072 ± 0.043 | 0.841 ± 0.151 | 0.968 ± 0.034 |
| Stock | Chronos | gaussian | 0.958 ± 0.037 | 0.990 ± 0.008 | 0.006 ± 0.007 | 0.175 ± 0.081 | 0.799 ± 0.197 | 0.974 ± 0.027 |
| Stock | Tirex | $\mathcal{W}_1$-ACAS | 0.955 ± 0.027 | 0.983 ± 0.008 | 0.010 ± 0.009 | 0.072 ± 0.043 | **0.984 ± 0.010** | 0.985 ± 0.024 |
| Stock | Tirex | conformal | 0.947 ± 0.039 | 0.985 ± 0.006 | 0.010 ± 0.007 | 0.079 ± 0.047 | 0.880 ± 0.104 | 0.987 ± 0.017 |
| Stock | Tirex | gaussian | 0.964 ± 0.034 | 0.989 ± 0.007 | 0.006 ± 0.005 | 0.179 ± 0.084 | 0.855 ± 0.133 | 0.986 ± 0.018 |
| Stock | TTM | $\mathcal{W}_1$-ACAS | 0.967 ± 0.028 | 0.989 ± 0.008 | 0.009 ± 0.011 | 0.074 ± 0.041 | 0.963 ± 0.053 | 0.991 ± 0.015 |
| Stock | TTM | conformal | 0.963 ± 0.030 | 0.989 ± 0.007 | 0.008 ± 0.009 | 0.071 ± 0.046 | 0.825 ± 0.184 | 0.975 ± 0.031 |
| Stock | TTM | gaussian | 0.965 ± 0.018 | 0.988 ± 0.005 | 0.004 ± 0.004 | 0.182 ± 0.087 | 0.818 ± 0.164 | 0.974 ± 0.028 |
| IOPS | - | KShapeAD | 0.365 ± 0.358 | 0.703 ± 0.064 | 0.699 ± 0.337 | 0.171 ± 0.149 | 0.025 ± 0.028 | 0.049 ± 0.044 |
| IOPS | - | POLY | 0.493 ± 0.410 | 0.854 ± 0.113 | 0.214 ± 0.318 | 0.442 ± 0.308 | 0.042 ± 0.067 | 0.230 ± 0.121 |
| IOPS | - | Sub-KNN | 0.334 ± 0.342 | 0.695 ± 0.038 | 0.693 ± 0.378 | 0.137 ± 0.187 | 0.021 ± 0.029 | 0.073 ± 0.095 |
| IOPS | - | Sub-PCA | 0.497 ± 0.432 | 0.780 ± 0.112 | 0.360 ± 0.341 | 0.352 ± 0.236 | 0.059 ± 0.070 | 0.206 ± 0.158 |
| IOPS | - | SAND | 0.052 ± 0.038 | 0.703 ± 0.039 | 0.808 ± 0.196 | 0.091 ± 0.056 | 0.008 ± 0.004 | 0.082 ± 0.055 |
| IOPS | - | CNN* | 0.865 ± 0.224 | 0.870 ± 0.091 | 0.018 ± 0.040 | 0.800 ± 0.271 | 0.102 ± 0.071 | 0.285 ± 0.165 |
| IOPS | - | OmniAnomaly* | 0.734 ± 0.275 | 0.803 ± 0.113 | 0.161 ± 0.261 | 0.639 ± 0.286 | 0.044 ± 0.038 | 0.207 ± 0.129 |
| IOPS | - | USAD* | 0.493 ± 0.348 | 0.771 ± 0.113 | 0.387 ± 0.314 | 0.402 ± 0.278 | 0.041 ± 0.044 | 0.130 ± 0.077 |
| IOPS | - | MOMENT_ZS | 0.565 ± 0.347 | 0.870 ± 0.098 | 0.074 ± 0.124 | 0.665 ± 0.293 | 0.052 ± 0.060 | **0.337 ± 0.261** |
| IOPS | Chronos | $\mathcal{W}_1$-ACAS | 0.886 ± 0.147 | 0.882 ± 0.062 | 0.004 ± 0.008 | **0.005 ± 0.014** | 0.158 ± 0.143 | 0.291 ± 0.151 |
| IOPS | Chronos | conformal | 0.850 ± 0.175 | 0.884 ± 0.067 | 0.008 ± 0.013 | 0.006 ± 0.013 | 0.151 ± 0.160 | 0.298 ± 0.208 |
| IOPS | Chronos | gaussian | 0.543 ± 0.217 | 0.811 ± 0.035 | 0.024 ± 0.019 | 0.021 ± 0.020 | 0.109 ± 0.087 | 0.308 ± 0.208 |
| IOPS | Tirex | $\mathcal{W}_1$-ACAS | **0.921 ± 0.073** | **0.889 ± 0.061** | **0.003 ± 0.007** | 0.007 ± 0.014 | **0.184 ± 0.144** | 0.296 ± 0.167 |
| IOPS | Tirex | conformal | 0.875 ± 0.126 | 0.888 ± 0.061 | 0.007 ± 0.013 | 0.006 ± 0.012 | 0.151 ± 0.140 | 0.301 ± 0.207 |
| IOPS | Tirex | gaussian | 0.528 ± 0.232 | 0.800 ± 0.035 | 0.026 ± 0.020 | 0.023 ± 0.019 | 0.113 ± 0.089 | 0.304 ± 0.207 |
| IOPS | TTM | $\mathcal{W}_1$-ACAS | 0.871 ± 0.203 | 0.870 ± 0.075 | 0.009 ± 0.029 | **0.005 ± 0.012** | 0.167 ± 0.136 | 0.304 ± 0.153 |
| IOPS | TTM | conformal | 0.826 ± 0.212 | 0.875 ± 0.084 | 0.019 ± 0.050 | 0.007 ± 0.013 | 0.123 ± 0.102 | 0.313 ± 0.224 |
| IOPS | TTM | gaussian | 0.558 ± 0.228 | 0.809 ± 0.045 | 0.032 ± 0.048 | 0.021 ± 0.024 | 0.109 ± 0.092 | 0.309 ± 0.215 |

Table 4: **Performance Summary per datasets.** Entries indicate the mean ± standard deviation computed by averaging within each dataset group. Higher numbers are better for PA-F1, Affiliation-F, AUC-PR, VUS-PR; lower numbers are better for FPR, and calibration error (CalErr). Methods marked with * denote deep learning semi-supervised approaches; the best overall method is shown in **bold**, and the best non–semi-supervised method is underlined when different from the bold one.

| Dataset Forecaster | IOPS | MSL | NAB | NEK | Stock | WSD | YAHOO |
|---|---|---|---|---|---|---|---|
| **MAE** | | | | | | | |
| Chronos | 1.50 ± 1.83 | 0.06 ± 0.06 | 244.96 ± 1108.10 | 0.38 ± 0.21 | 6.81 ± 3.89 | 149.73 ± 200.81 | 280.75 ± 232.69 |
| TiRex | 1.45 ± 1.77 | 0.06 ± 0.05 | 234.95 ± 1063.39 | 0.37 ± 0.22 | 6.85 ± 4.00 | 138.51 ± 182.07 | 250.80 ± 220.62 |
| TTM | 1.45 ± 1.74 | 0.09 ± 0.07 | 267.97 ± 1219.00 | 0.59 ± 0.41 | 7.59 ± 4.74 | 139.11 ± 183.01 | 481.91 ± 252.17 |
| **RMSE** | | | | | | | |
| Chronos | 2.38 ± 2.87 | 0.20 ± 0.19 | 341.38 ± 1487.54 | 0.75 ± 0.38 | 15.33 ± 9.43 | 212.32 ± 286.53 | 461.99 ± 433.56 |
| TiRex | 2.31 ± 2.78 | 0.20 ± 0.19 | 326.81 ± 1426.95 | 0.75 ± 0.39 | 15.40 ± 9.53 | 198.23 ± 261.90 | 417.93 ± 411.42 |
| TTM | 2.30 ± 2.74 | 0.22 ± 0.21 | 373.15 ± 1654.63 | 0.94 ± 0.54 | 15.37 ± 9.53 | 196.56 ± 262.49 | 683.50 ± 429.53 |

Table 5: **TSFM Forecasting Performance (MAE and RMSE) per Multivariate Dataset.** Mean Absolute Error and Root Mean Squared Error for each TSFM model on the anomaly detection datasets, computed using a 15-step-ahead forecast and a context length of 52 past observations. Entries report mean ± standard deviation across all series within each dataset. Overall, forecasting performance is similar across models; the slightly higher error of TTM on YAHOO aligns with its correspondingly lower AD performance in Table 2.

| Dataset | Forecaster | AD Model | PA-F1 ↑ | Affiliation-F ↑ | FPR ↓ | CalErr ↓ | AUC-PR ↑ | VUC-PR ↑ |
|---|---|---|---|---|---|---|---|---|
| TAO | - | CNN* | 0.998 ± 0.001 | 0.999 ± 0.000 | **0.000 ± 0.000** | 0.612 ± 0.044 | 0.895 ± 0.094 | 0.999 ± 0.001 |
| TAO | - | OmniAnomaly* | 0.377 ± 0.021 | 0.863 ± 0.053 | 0.321 ± 0.153 | 0.497 ± 0.136 | 0.311 ± 0.039 | 0.940 ± 0.051 |
| TAO | - | USAD* | 0.172 ± 0.061 | 0.679 ± 0.006 | 0.986 ± 0.018 | 0.033 ± 0.027 | 0.018 ± 0.005 | 0.097 ± 0.017 |
| TAO | Chronos_allctx | $\mathcal{W}_1$-ACAS-F | **1.000 ± 0.000** | **1.000 ± 0.000** | **0.000 ± 0.000** | 0.029 ± 0.028 | 0.901 ± 0.085 | 0.998 ± 0.003 |
| TAO | Chronos_allctx | $\mathcal{W}_1$-ACAS-H | 0.999 ± 0.001 | **1.000 ± 0.000** | **0.000 ± 0.000** | 0.251 ± 0.110 | **0.945 ± 0.047** | 0.999 ± 0.001 |
| TAO | Tirex_allctx | $\mathcal{W}_1$-ACAS-F | **1.000 ± 0.000** | **1.000 ± 0.000** | **0.000 ± 0.000** | **0.026 ± 0.023** | 0.907 ± 0.100 | **1.000 ± 0.001** |
| TAO | Tirex_allctx | $\mathcal{W}_1$-ACAS-H | 0.999 ± 0.000 | **1.000 ± 0.000** | **0.000 ± 0.000** | 0.257 ± 0.109 | 0.938 ± 0.060 | 0.998 ± 0.003 |
| GECCO | - | CNN* | 0.583 | 0.875 | 0.139 | 0.860 | **0.294** | 0.152 |
| GECCO | - | OmniAnomaly* | 0.579 | 0.840 | 0.206 | 0.792 | 0.216 | 0.186 |
| GECCO | - | USAD* | 0.561 | 0.772 | 0.353 | 0.643 | 0.038 | 0.091 |
| GECCO | Chronos-allctx | $\mathcal{W}_1$-ACAS-F | 0.704 | **0.882** | **0.030** | **0.014** | 0.231 | 0.236 |
| GECCO | Chronos-allctx | $\mathcal{W}_1$-ACAS-H | 0.606 | 0.835 | 0.044 | 0.316 | 0.110 | 0.127 |
| GECCO | Tirex-allctx | $\mathcal{W}_1$-ACAS-F | **0.742** | 0.864 | 0.034 | 0.026 | 0.237 | **0.241** |
| GECCO | Tirex-allctx | $\mathcal{W}_1$-ACAS-H | 0.587 | 0.842 | 0.039 | 0.320 | 0.124 | 0.136 |
| LTDB | - | CNN* | 0.910 ± 0.137 | **0.837 ± 0.105** | 0.228 ± 0.404 | 0.657 ± 0.368 | **0.247 ± 0.191** | 0.343 ± 0.272 |
| LTDB | - | OmniAnomaly* | 0.875 ± 0.184 | 0.830 ± 0.106 | 0.297 ± 0.471 | 0.511 ± 0.445 | 0.211 ± 0.176 | 0.272 ± 0.204 |
| LTDB | - | USAD* | 0.639 ± 0.380 | 0.866 ± 0.126 | 0.280 ± 0.481 | 0.540 ± 0.363 | 0.178 ± 0.277 | **0.453 ± 0.385** |
| LTDB | Chronos-allctx | $\mathcal{W}_1$-ACAS-F | 0.845 ± 0.142 | 0.792 ± 0.056 | 0.065 ± 0.054 | 0.038 ± 0.039 | 0.227 ± 0.161 | 0.265 ± 0.187 |
| LTDB | Chronos-allctx | $\mathcal{W}_1$-ACAS-H | 0.910 ± 0.054 | 0.816 ± 0.024 | 0.068 ± 0.070 | 0.162 ± 0.119 | 0.170 ± 0.116 | 0.263 ± 0.185 |
| LTDB | Tirex-allctx | $\mathcal{W}_1$-ACAS-F | 0.888 ± 0.101 | 0.814 ± 0.096 | **0.047 ± 0.050** | 0.026 ± 0.029 | 0.227 ± 0.157 | 0.260 ± 0.183 |
| LTDB | Tirex-allctx | $\mathcal{W}_1$-ACAS-H | **0.937 ± 0.041** | 0.829 ± 0.067 | 0.060 ± 0.067 | 0.134 ± 0.108 | 0.171 ± 0.114 | 0.263 ± 0.181 |
| Genesis | - | CNN* | 0.649 | 0.852 | 0.004 | 0.896 | 0.031 | 0.045 |
| Genesis | - | OmniAnomaly* | 0.473 | 0.873 | 0.009 | 0.784 | 0.018 | 0.028 |
| Genesis | - | USAD* | 0.094 | 0.864 | 0.088 | 0.272 | 0.031 | **0.074** |
| Genesis | Chronos-allctx | $\mathcal{W}_1$-ACAS-F | 0.793 | 0.891 | **0.002** | 0.001 | 0.075 | 0.066 |
| Genesis | Chronos-allctx | $\mathcal{W}_1$-ACAS-H | 0.850 | 0.903 | **0.002** | 0.058 | **0.084** | 0.066 |
| Genesis | Tirex-allctx | $\mathcal{W}_1$-ACAS-F | 0.850 | **0.912** | 0.002 | **0.000** | 0.039 | 0.029 |
| Genesis | Tirex-allctx | $\mathcal{W}_1$-ACAS-H | **0.873** | 0.897 | 0.004 | 0.116 | 0.058 | 0.046 |

Table 6: **Performance Summary per Multivariate Dataset.** Entries indicate the mean ± standard deviation computed by averaging within each dataset group. Higher numbers are better for PA-F1, Affiliation-F, AUC-PR, VUS-PR; lower numbers are better for FPR, and calibration error (CalErr). Standard deviations are omitted for GECCO and Genesis, as each corresponds to a single multivariate time series (9 features, 138k samples for GECCO; 18 features, 16k samples for Genesis). Methods marked with * denote deep learning semi-supervised approaches; the best overall method is shown in **bold**.