# OpenReview forum: "Adaptive Conformal Anomaly Detection with Time Series Foundation Models for Signal Monitoring."
_ICLR.cc/2026/Conference — ICLR 2026 Poster_

### Official Review · Reviewer_2NiA · 2025-10-31

**Soundness:** 3
**Presentation:** 3
**Contribution:** 2
**Rating:** 4
**Confidence:** 4

**Summary:**

This paper proposes a post-hoc, model-agnostic adaptive conformal anomaly detection framework that leverages pre-trained Time Series Foundation Models without additional fine-tuning.  Experiments on a total of seven benchmark datasets exhibit consistent superior performace over existing methods.

**Strengths:**

The paper focuses on key pain points in industrial time series anomaly detection such as limited high-quality data, lack of training expertise, and non-stationary data distributions. These are critical challenges in fields like predictive maintenance.  Experiments are comprehensive and well-designed that cover a total of seven benchmark datasets.

**Weaknesses:**

One caveat is that this paper focuses exclusively on univariate time series anomaly detection, without extending it to multivariate time series scenarios, despite multivariate time series are common in industrial monitoring and predictive maintenance.  In addition, it provides no details on computational overhead or scalability of the proposed method. For practical scenarios requiring real-time monitoring, computational costs are critical and could limit practical deployments.

**Questions:**

Can the proposed method be extended to deal with multi-dimensional time series data?

---

> ### Author Response · Authors · 2025-11-27
>
> We thank the reviewer for the helpful feedback, which strengthened our work. We updated the manuscript (new additions in blue) and address the weaknesses and questions below.
>
> **W1 and Q1: Multivariate Extension**
>
> We thank the reviewer for highlighting this point. In Appendix C.2.4, we added a subsection showing that the method extends naturally to the multivariate setting by applying standard p-value aggregation methods (Fisher, 1970; Heard & Rubin-Delanchy (2018); Wilson, 2019) across dimensions. We also included experiments on four multivariate time series anomaly detection datasets (Table 6), demonstrating that combining TSFM zero-shot forecasters with $\mathcal{W}_1$-ACAS and p-value aggregation achieves top or highly competitive performance compared with strong semi-supervised baselines.
>
> **W2. Missing Analysis of Computational Cost**
>
> The average per-sample computation time of $\mathcal{W}_1$-ACAS with a 15-step forecast is 0.025 ± 0.012 seconds per sample per feature on a single V100 32 GB GPU. Note that this implementation updates weights for all 15 predictors serially, these updates are independent and can be parallelized to further reduce runtime. We clarified this in "Computation Time" paragraph in Appendix C.2.3.

---

### Official Review · Reviewer_n5Nx · 2025-11-01

**Soundness:** 3
**Presentation:** 3
**Contribution:** 3
**Rating:** 6
**Confidence:** 3

**Summary:**

This paper presents **W1-ACAS** (1-Wasserstein Adaptive Conformal Anomaly Score), a post-hoc adaptive conformal anomaly detection method for monitoring time series using pre-trained foundation models without requiring fine-tuning. The key contributions include: (1) producing interpretable anomaly scores directly linked to false alarm rates (p-values), enabling transparent threshold selection; (2) using weighted quantile conformal prediction that adapts online to distribution shifts by learning optimal weights via Wasserstein distance minimization; (3) providing a model-agnostic solution that works seamlessly with any time series foundation model while preserving theoretical guarantees; and (4) addressing practical industrial challenges such as limited data availability, lack of training expertise, and need for immediate deployment. Experiments on synthetic and real-world benchmark datasets demonstrate that W1-ACAS achieves strong detection performance with better calibration and lower false positive rates compared to baseline methods, while maintaining robustness to temporal distribution shifts.

**Strengths:**

1. Interesting idea of improving TSFMs' performances on anomaly detection (even if the TSFMs are trained for forecasting)
2. Solid mathematical derivations

**Weaknesses:**

1. Missing LLM usage section
2. The baselines mainly consist of TSFM-based baselines. Some of these TSFMs are not specifically trained for anomaly detection, so it's easy to improve their performances.
3. Should also include more recent (after 2023) deep-learning based baseline.
4. Improvement not that impressive give the high standard deviation

**Questions:**

1. What is the fundamental difference between this paper and [1]
2. Does this also work on fine-tuned non-FM anomaly detection model?
3. The TSFMs are forecasting models. They are not idea for contextual anomalies. Does this method help to alleviate that disadvantage?
4. Is this applicable to reconstruction-based models.




[1]  Margaux Zaffran, Olivier Féron, Yannig Goude, Julie Josse, and Aymeric Dieuleveut. Adaptive conformal predictions for time series. In Proceedings of the ICML. PMLR, 2022.

---

> ### Author Response · Authors · 2025-11-27
>
> We thank the reviewer for the valuable feedback, which significantly helped improve the manuscript. We updated the manuscript (new additions in blue) and address the weaknesses and questions below.
>
> **W1. Missing LLM usage section**
>
> The required LLM usage disclosure is already included in Appendix D: “The Use of Large Language Models (LLMs)”
>
> **W2–W3. Regarding TSFM baselines and the need for deep-learning based baselines**
>
> The TSFM-based zero-shot baselines (TSFM–Gaussian) are included to illustrate how a standard/basic forecasting-error approach performs and how $\mathcal{W}_1$-ACAS can leverage and enhance a zero-shot TSFM forecaster using only online updates (without labeled data) while producing calibrated anomaly scores and low false-alarm rates.
>
> In the original manuscript we included the top-performing unsupervised anomaly detection methods from the recent benchmark of Liu & Paparrizos (2024). Following the reviewer’s recommendation, we expanded our baselines to incorporate additional strong deep learning anomaly detection methods listed in Audibert et al. (2022) and Liu & Paparrizos (2024). Tables 1–4 now report results for CNN (Munir et al., 2018), USAD (Audibert et al., 2020), and OmniAnomaly (Su et al., 2019), which are top semi-supervised performers in these benchmarks. Although these approaches are trained on the non-anomalous portion of each dataset, our method remains competitive. We also added MOMENT (Goswami et al., 2024), a general-purpose TSFM that provides zero-shot anomaly scoring. Details of these additions appear in Section 6 and Appendix C.2.1.
>
> We further extended the evaluation to multivariate settings. Appendix C.2.4 describes how our method naturally generalizes via $p$-value aggregation, and Table 6 reports performance on 4 multivariate datasets.
>
> **W4. Improvement not that impressive give the high standard deviation**
>
> Tables 2–4 in the Appendix highlight per-dataset results, showing that the improvements are consistent at the dataset level and not only in aggregated summaries. In many datasets, the gains over using a standard TSFM forecasting-error score with a Gaussian assumption are substantial, even when variance across series is high (e.g., NEK, WSD, IOPS). This complements the aggregated results and clarifies where the improvements are most pronounced.
>
> **Q1. Difference from Zaffran et al. (ICML 2022)**
>
> Zaffran et al. focus on calibrating a single predictive quantile for a fixed target coverage level by optimizing a pinball-loss objective. In contrast, our method uses a Wasserstein-1–based adaptive estimator that adjusts the entire forecast-error distribution, implicitly calibrating all quantile levels simultaneously. This yields a full p-value mapping rather than calibrating a single quantile.
>
> **Q2. Does this also work on fine-tuned non-FM anomaly detection model?**
>
> Yes. The approach is model-agnostic and can be applied to any anomaly detection method (foundation model or not) as long as it outputs a scalar anomaly score. $\mathcal{W}_1$-ACAS operates entirely on the score distribution, so no assumptions are made about the underlying model. If multiple scores are available, they can be combined at the p-value level, since the method normalizes all scores through the conformal calibration step.
>
> **Q3. The TSFMs are forecasting models. They are not idea for contextual anomalies. Does this method help to alleviate that disadvantage?**
>
> If the observed error is marginally within expected values but conditionally inconsistent with its recent temporal pattern, the adaptive p-value mapping will flag it. The method would capture contextual anomalies through deviations in the local, time-dependent error distribution, rather than relying solely on marginal error magnitude.
>
> **Q4. Is this applicable to reconstruction-based models.**
> Yes. The reconstruction error produced by any reconstruction-based model can be used directly as the nonconformity score. $\mathcal{W}_1$-ACAS operates on the score distribution and can therefore calibrate reconstruction errors in exactly the same way as forecast-error–based scores.

---

### Official Review · Reviewer_dKjt · 2025-11-01

**Soundness:** 3
**Presentation:** 2
**Contribution:** 3
**Rating:** 6
**Confidence:** 3

**Summary:**

This work introduces a post-hoc adaptive conformal anomaly detection method for time series data, while combining predictions from pre-trained foundation models without fine-tuning.

**Strengths:**

- Sound approach that builds on the conformal prediction for non-exchangeable scenarios formulation (Barber, 2023) for a concrete application in time series anomaly detection
- A good number of datasets considered in the evaluation
- Promising results

**Weaknesses:**

Within the baselines, the paper could consider a broader number of competitors, encompassing classical and deep learning based. All the considered baselines report very poor results, although there are well-established and well-performing AD methods in the literature. One gets the impression that the evaluation is not fully fair. The authors may consider [1] for a starting point.

[1] https://doi.org/10.1016/j.patcog.2022.108945

**Questions:**

- Table 1 provides a summary of the performance of the proposed method over all the datasets. While the full table appears in the appendix, it would be helpful to include a per-dataset analysis in the main text to provide readers with insights into which scenarios the proposed method performs best/worst.
- It is surprising to see that the baselines have a drop in performance when using point-adjusted F1. Usually, this is not the case. Methods tend to have a higher PA-F1 than a standard one. Could you explain?
- From the results, it seems there is no clear best foundation model to be coupled with the proposed method. It would be nice to discuss this. Where do the differences in performance metrics stem from? A per-dataset analysis, as previously suggested, could provide insights.
- How sensitive is the proposed method to the different hyperparameters that need to be adjusted?

---

> ### Author Response · Authors · 2025-11-27
>
> We thank the reviewer for the comments, which substantially improved our analysis, manuscript, and references. We updated the manuscript (new additions in blue) and address the weaknesses and questions below.
>
> **W1. More Baselines**
>
> We expanded our baseline to include several strong deep learning anomaly detection methods, as recommended. Tables 1–4 now report results for CNN (Munir et al., 2018),  USAD (Audibert et al., 2020), and OmniAnomaly (Su et al., 2019), which are top-performing semi-supervised approaches in Liu & Paparrizos (2024) and listed in Audibert et al. (2022). Although these models are trained on the non-anomalous portion of each dataset, our method remains competitive. We also added MOMENT (Goswami et al., 2024), a general-purpose TSFM that offers zero-shot anomaly scoring. Details of these additions are provided in Section 6 and Appendix C.2.1.
>
> We further extended the evaluation to multivariate settings. Appendix C.2.4 describes how our method naturally generalizes via $p$-value aggregation, and Table 6 reports performance on 4 multivariate datasets.
>
> **Q1. Need for Per-Dataset Analysis in Main Text**
>
> We added a new Figure 2 in Section 6 that presents heatmaps of the average performance per dataset for a representative subset of top-performing methods across multiple metrics. This provides a clear view of where each method excels or struggles. As shown, our approach combined with TSFM forecasters performs strongly across datasets, including when compared against the best semi-supervised competitor (CNN).
>
> **Q2. Unexpected Drop in PA-F1 for Baselines**
>
> Please note that in our analysis we are not reporting standard F1, we only report Point-Adjusted F1 and Affiliation-F. The observed behavior (PA-F1 < Affiliation-F for the baselines) is consistent with Table 5 in Liu & Paparrizos (2024). As shown in their Fig. 5 (e.g., S7, S9), PA-F1 is not necessarily higher than Affiliation-F. If we are misunderstanding the reviewer’s question, we would be happy to clarify further.
>
> **Q3. Differences Across TSFM Models**
>
> Tirex and Chronos are pretrained on closely related time-series corpora, which likely explains why their forecast-error distributions are often similar across datasets (see new added Table 5 which summarizes their forecasting performance per dataset). Since $\mathcal{W}_1$-ACAS detects anomalies through local shifts in these errors, comparable forecast behavior may lead to comparable anomaly-detection performance. For instance TTM presents slightly higher forecasting error on YAHOO and a poorer anomaly-detection performance on it. This is further discussed in "Per-dataset performance" paragraph in Appendix C.2.3.
>
> **Q4. Hyperparameter Sensitivity**
>
> We added a hyperparameter-sensitivity analysis in Appendix C.2.3. In addition to the sweep over forecast horizons (Fig. 8), we now include Figures 9–11, which vary the initial learning rate $\gamma$, the batch size $n_b$ used in the Wasserstein distance, and the critical alarm rate $\alpha_c$ (which controls the maximum $p$-value resolution). These are the main tunable parameters in our algorithm. Overall, the method is not highly sensitive to these choices: small initial learning rates (we use Adam updates) and small $n_b$ values work well in practice, while $\alpha_c$ primarily determines how many in-distribution samples are needed. When the score distribution changes rapidly, using a smaller $n_b$ is preferable.

---

> > ### Comment · Reviewer_dKjt · 2025-11-27
> >
> > Thanks for your answers. My impression of the work remains positive and I think the authors did a good job in the rebuttal.

---

### Author Response · Authors · 2025-11-27
**Thanks to Reviewers and Summary of Main Updates**

We thank all reviewers for the time dedicated to evaluating our work and for the constructive feedback and valuable references, which have substantially improved the quality of the manuscript.

This work aims to demonstrate how the proposed $\mathcal{W}_1$-ACAS framework can leverage and enhance a zero-shot TSFM forecaster using only online updates based on past observations (without labeled data), while producing interpretable, calibrated anomaly scores with low false-alarm rates.

We have revised the manuscript accordingly, highlighting in blue the major additions across both the main text and the Appendix. Below we summarize the current key updates:

1.	**Expanded Benchmark: Four Additional Deep Learning Baselines.**
Following the reviewers' recommendations, we incorporated four strong baselines: CNN (Munir et al., 2018), USAD (Audibert et al., 2020), OmniAnomaly (Su et al., 2019) which are semi-supervised deep learning baselines, and MOMENT (Goswami et al., 2024), a general-purpose TSFM that supports zero-shot anomaly scoring. Tables 1–4, Section 6, and Appendix C.2.1 have been updated with these models. We also added a new Figure 2 showing dataset-level heatmaps for a representative subset of top-performing methods across multiple metrics, offering a clear view of where each method excels or underperforms. Our method remains competitive even against semi-supervised baselines trained on non-anomalous data.

2.	**Hyperparameter Sensitivity Analysis.**
We expanded the sensitivity study in Appendix C.2.3. In addition to the forecast horizon sweep (Fig. 8), we now include Figures 9–11 analyzing the effect of the initial learning rate $\gamma$, the batch size $n_b$ used in Wasserstein updates, and the critical alarm rate $\alpha_c$ (which governs the maximum p-value resolution) showing robustness to a variety of configurations.

3.	**Multivariate Time Series Anomaly Detection.**
Appendix C.2.4 now describes how $\mathcal{W}_1$-ACAS easily extends to the multivariate setting via standard p-value aggregation (Fisher, 1970; Heard & Rubin-Delanchy (2018); Wilson, 2019) across dimensions. We also added experiments on four multivariate TSAD datasets (Table 6) where $\mathcal{W}_1$-ACAS achieves top or highly competitive performance compared with strong semi-supervised baselines.

4.	**Additional Analysis and Discussion.**
In Appendix C.2.3 (“Per-dataset performance”), we added Table 5 summarizing TSFM forecasting performance per dataset and discussing its relationship to anomaly-detection performance. We also added a “Computation Time” paragraph reporting average update times and outlining potential parallelization strategies.

---

### Public Comment · ~Natalia_Martinez1 · 2026-04-22
**Typo correction in Eq. 5 of Proposition 3.1**

Hi,
We would like to flag a typo in the uploaded version of our paper. In Eq. 5 of Proposition 3.1, the $+$ signs in the second inequality should be $-$, following Theorem 3 in Barber et al. (2023). Then the correct expression for Eq 5 is:

$\mathbb{P}(A_{n+1} = 1) \leq \alpha +  \sum^n_{i=1}\frac{w_i}{||\mathbf{w}||_1 + 1} dTV (\mathbf{s},\mathbf{s}^i) $

$\mathbb{P}(A_{n+1} = 1) > \alpha - \sum^n_{i=1} \frac{w_i}{||\mathbf{w}||_1 + 1} dTV(\mathbf{s},{\mathbf{s}}^i) - \frac{1}{||\mathbf{w}||_1 + 1} $

This is a typographical error only and does not affect the validity of the proposition or subsequent results. The corrected version will be available on arXiv. We apologize for the confusion.

UPDATE: Typo fixed in camera ready revision.

---

### Meta-Review · Area_Chair_Ku16 · 2026-01-06

**Summary:**

The reviewers raised concerns about the fairness and completeness of the experimental evaluation, including the lack of strong deep learning baselines, limited per-dataset analysis, missing hyperparameter sensitivity studies, unclear applicability to multivariate time series, and insufficient discussion of computational cost. There were also questions about how the method differs from prior adaptive conformal prediction work and whether it generalizes beyond forecasting-based foundation models.

 The authors addressed these concerns largely in the rebuttal and revised manuscript. They added multiple strong baselines, extensive sensitivity analyses, multivariate experiments, per-dataset results, and clear clarifications of methodological novelty and scope. One reviewer acknowledge the feedback and remain positive.

**Reviewer Concerns:**

Concerns addressed by the rebuttal:
- Added strong deep learning baselines and a recent TSFM baseline.
- Included per-dataset analysis and clearer discussion of performance variation.
- Added extensive hyperparameter sensitivity analysis.
- Extended the method to multivariate time series with new experiments.
- Clarified computational cost and scalability.
- Clearly distinguished the method from prior adaptive conformal prediction work.
- Clarified applicability to non-foundation and reconstruction-based models.

Remaining minor concerns:
- Performance gains are sometimes modest on aggregated metrics.
- Some variance remains high across datasets.

**Reviewer Scores:**

Reviewers dKjt and n5Nx provide ratings of 6, which are positive already.

Reviewer 2NiA has a rating of 4. His rating is likely to increase, as both multivariate extension and computational cost concerns were fully addressed.

---

### Decision · Program_Chairs · 2026-01-26

Accept (Poster)